# LLM4Solver: Large Language Model for Efficient Algorithm Design of Combinatorial Optimization Solver

## Abstract

The optimization of algorithms in exact combinatorial optimization (CO) solver plays a fundamental role in operations research. However, due to the extensive requirements on domain knowledge and the large search space for algorithm design, the refinement on these algorithms remains highly challenging for both manual and learning-based paradigms. To tackle this problem, we propose a novel machine learning framework—large language model for exact combinatorial optimization solver (LLM4Solver)—to *efficiently* design high-quality algorithms of the CO solvers. The core idea is that, instead of searching in the high-dimensional and discrete symbolic space from scratch, we can utilize the prior knowledge learned from large language models to directly search in the space of programming languages. Specifically, we first use a pre-trained LLM as the generator for high-quality algorithms. Then, to efficiently explore the discrete and non-gradient algorithm space, we employ a derivative-free evolutionary framework as the algorithm optimizer. Experiments on extensive benchmarks show that the algorithms learned by LLM4Solver *significantly* outperform all the state-of-the-art (SOTA) human-designed and learning-based policies (on GPU) in terms of the solution quality, the solving efficiency, and the cross-benchmark generalization ability. The appealing features of LLM4Solver include 1) the high training efficiency to outperform SOTA methods within ten iterations, and 2) the high cross-benchmark generalization ability on heterogeneous MIPLIB 2017. LLM4Solver shows the encouraging potential to efficiently design algorithms for the next generation of modern CO solvers.

## 1 Introduction

Combinatorial optimization (CO), which aims to find an optimal object from a finite solution set, is one of the most fundamental models in operations research (OR) (Achterberg, 2007; Bengio et al., 2021). It is widely used to formulate a series of important real-world tasks, e.g., scheduling, transportation, and management (Liu et al., 2008; Chen, 2010; Ma et al., 2019; Paschos, 2014). In these applications, the solving efficiency and the solution quality are usually related to enormous economic value (Kuang et al., 2023; Achterberg, 2007). Thus, the optimization for algorithms on exact CO solvers plays a fundamental role in the field of OR. Popular exact CO solvers like SCIP (Gleixner et al., 2018) and Gurobi (Gurobi Optimization, LLC, 2023) employ a rich set of hard-coded heuristics, whose efficacy directly affects the performance of the CO solvers. Due to the complexity of these heuristics, designing and optimizing them typically demand substantial domain expertise, significant manual adjustments, and intricate workflows(Achterberg, 2007; Bengio et al., 2021).

Recently, there has been an explosive surge in the use of machine learning (ML) techniques to enhance exact CO solvers. These learning-based approaches can be roughly divided into two classes. One class incorporates deep neural networks (DNNs) to approximate different components in CO solvers, e.g., the branching (Gasse et al., 2019), the cut selection (Wang et al., 2023), and the primal heuristics (Paulus & Krause, 2024; Nair et al., 2020). Note that these DNN models fail to explain what patterns they have learned that accelerate the CO solvers (Kuang et al., 2024a), and thus they fail to help researchers further optimize the human-designed heuristics in solvers. To tackle this problem, the other class (Kuang et al., 2024a;b) employs symbolic discovery approaches to learn more interpretable heuristics. Currently, these approaches are proved effective mainly on the branching

component (Achterberg, 2007), in which symbolic regression (SR) is employed to learn interpretable scoring functions that outperform DNN policies on purely CPU-based devices.

Previous symbolic discovery approaches(Kuang et al., 2024a;b) , though effective for branching, have two general limitations that severely hinder their potential applications to CO solvers. (1) Due to the high-dimensional and discrete search space for symbolic discovery and its nature of searching from scratch, existing SR-based approaches suffer from high computational costs. (2) The vastly different intrinsic nature of CO problems across various scenarios causes most approaches to fail in generalizing to various benchmarks. However, we hope that learning-based approaches will design more generic heuristics like human-designed ones to enhance the built-in performance of the solvers.

The powerful capabilities of the large language models (LLM) in text comprehension and logic generation have attracted widespread attention(Naveed et al., 2023; Yang et al., 2023), offering new approaches for algorithm design. Romera-Paredes et al. (2024) combine LLM with the island-based evolution for mathematical discovery. Liu et al. (2024) and Ye et al. (2024) leverage LLM to revise heuristics for online bin packing, traveling salesman problems, and flow shop scheduling problems. Sun et al. (2024) create a multi-agent-based framework to improve the heuristics of SAT problems. These works have demonstrated impressive results in scenarios like mathematical discovery, and designing heuristics for classical CO and SAT problems. However, while online bin packing and traveling salesman problems are highly representative and important CO problems, when modeled and solved using general MILP formulations in CO solvers, it is critical to investigate heuristics to find feasible solutions in more general MILP problems. Currently, there are no LLM-based heuristic optimization methods specifically designed for general MILP problems and research on such methods holds more generic scientific significance.

In this work, we propose an automatic algorithm design framework—large language model for exact combinatorial optimization solver (LLM4Solver)—to *efficiently* design high-quality diving heuristics of the CO solvers. Specifically, we first use the LLM as an algorithm generator, leveraging it to design three operators: initialization, crossover, and mutation, to generate new executable algorithm code with LLM's prior knowledge. Then, we treat the derivative-free evolutionary framework as an optimizer, utilizing it to iteratively optimize in the non-gradient algorithm space. Finally, we extend this framework through *multi-objective evolution* to leverage the heterogeneous characteristics of different CO problems to design more generic algorithms. Extensive experiments show that LLM4Solver-designed *interpretable* diving heuristics *significantly* outperform all the state-of-the-art (SOTA) human-designed and learning-based policies (on GPU) in terms of solution quality, solving efficiency, and cross-benchmark generalization ability.

We summarize the highly appealing features of LLM4Solver as follows. (1) High performance. LLM4Solver outperforms *all* the baselines, including both the human-designed diving heuristics in SCIP (Gleixner et al., 2018) and the SOTA learning-based policy on GPU (Paulus & Krause, 2024), in terms of the solution quality (Table 1) and the solving efficiency (Table 2). (2) Efficient searching. LLM4Solver outperforms the SOTA learning-based approaches within *only* four iterations and converges to optimum within ten iterations, respectively (Figure 2). (3) Strong cross-benchmark generalization ability. LLM4Solver can design a generic diving heuristic with high *cross-benchmark* generalization ability on different benchmarks (Table 3), including the highly challenging heterogeneous MIPLIB 2017 (Table 4). (4) Good interpretability. The programs with comments designed by LLM4Solver (Figure 4) clearly illustrate the execution logic of the algorithms, offering better interpretability compared to neural network parameters(Paulus & Krause, 2024) and purely symbolic expressions(Kuang et al., 2024a). LLM4Solver shows the potential to efficiently design high-quality and generic algorithms for the next generation of solvers, thereby enhancing their built-in capabilities.

## 2 PRELIMINARIES

### 2.1 EXACT COMBINATORIAL OPTIMIZATION SOLVERS AND DIVING HEURISTIC

In real-world scenarios, a series of CO problems can be modeled as Mixed Integer Linear Programmings (MILPs), taking the form of:

$$\arg\min_{\mathbf{x}}\{\mathbf{c}^\top \mathbf{x} | \mathbf{A}\mathbf{x} \le \mathbf{b}, \mathbf{l} \le \mathbf{x} \le \mathbf{u}, x_j \in \mathbb{Z} \,\forall j \in \mathcal{I}\},$$

where $\mathbf{c}$ denotes the objective coefficient vector, $\mathbf{A}$ the constraint matrix, $\mathbf{b}$ the constraint right hand side vector, $\mathbf{l}, \mathbf{u}$ respectively the lower and upper bounds and $\mathcal{I}$ denotes the index of integer variables. In exact solvers like SCIP (Gleixner et al., 2018), MILPs are solved with the branch-and-bound (B&B) algorithm. B&B recursively builds a search tree and expands the tree by selecting a variable $x_i$ to partition the problem into two subproblems. Specifically, one adds constraint $x_i \leq \lfloor x_i^* \rfloor$ and the other adds $x_i \geq \lceil x_i^* \rceil$, where the $x_i^*$ is the fractional value in the solution of the linear programming (LP) relaxation problem. Here, the LP relaxation problem is defined as $\arg\min_x \{\mathbf{c}^\top \mathbf{x} | \mathbf{A}\mathbf{x} \leq \mathbf{b}, x \in \mathbb{R}^n\}$
and its constraints are defined as $P^*$. Furthermore, B&B uses objective bounds to prune the tree and direct the exploration. Primal heuristics help solvers obtain stronger primal bounds and improve solving efficiency. Among them, diving heuristics is one of the most common primal heuristics and has a significant impact on the performance of solvers. They perform a depth-first search by iteratively rounding a variable and solving the modified LP relaxation problems until a feasible solution is found or infeasibility is proven. Algorithm 1 details the diving heuristic process. The scoring function $s$, used to select the variables and rounding direction, is the most crucial part of diving heuristics.

## 2.2 EVOLUTIONARY ALGORITHMS

Given a minimization problem $\arg\min_{v \in \mathcal{V}} h(v)$, evolutionary algorithms (EA)(Zhou et al., 2019) take $v$ as an individual and use *parent selection*, *crossover*, *mutation*, *fitness measure* and *survivor selection* operators to get better individuals, see Figure 1. After generations of iteration, EA outputs a population of feasible solutions to the problem.

Multi-objective evolutionary algorithms (MOEAs)(Deb et al., 2002; Zhang & Li, 2007) can implent on multi-objective minimization problem $\arg\min_{v \in \mathcal{V}}(h_1(v), h_2(v), ..., h_m(v))$. For two solutions $v, v'$ in a multi-objective minimization problem, we define that

- $v$ *weakly dominates* $v'$ (denoted as $v \preceq v'$) **iff.** $\forall 1 \leq i \leq m, h_i(v) \leq h_i(v')$.
- $v$ *dominates* $v'$ (denoted as $v \prec v'$) **iff.** $v \preceq v'$ and $\exists 1 \leq i \leq m, h_i(v) < h_i(v')$.

A feasible solution that any other solution cannot dominate is called the *Pareto optimal solution*. The set of all Pareto optimal solutions is called the *Pareto Front*. MOEAs leverage the evolution framework and output the Pareto Front of the multi-objective optimization problem.

## 2.3 PERFORMANCE MEASUREMENT

It is common to measure the *primal-dual gap* as the solving performance, taking the form as:

$$\gamma_{pd}(\tilde{z}, \tilde{z}^*) = \begin{cases} \frac{|\tilde{z} - \tilde{z}^*|}{\max(|\tilde{z}|, |\tilde{z}^*|)}, & if \ 0 < \tilde{z}\tilde{z}^* < \infty, \\ 1, & else, \end{cases}$$

where $\tilde{z}$ is the primal bound given by the incumbent feasible solution $\tilde{x}$ and $\tilde{z}^*$ is the dual bound given by the optimal solution of LP relaxation problem.

**Primal-dual integral** Considering that the primal-dual gap is subject to the final solution and the time limit setting, a more intuitive way is measuring the variation of the primal-dual gap during the solving process, i.e. calculating the *primal-dual integral* along the time steps:

$$PD(T) = \int_{t=0}^{T} \gamma_{pd}(\tilde{z}_t, \tilde{z}_t^*) dt.$$

**Primal gap** As the diving heuristics only aim to improve the primal performance, there is a necessity to introduce the *relative primal gap* to assess the effectiveness of diving heuristics, which is given by:

$$\gamma_p(\tilde{z}) = \frac{|\tilde{z} - z^\dagger|}{|z^\dagger|},$$

where $z^\dagger$ is the objective value of the optimal solution presolved. The primal gap intuitively shows the objective value distance between the current feasible solution and the global optimal solution. If $|z^\dagger| = 0$, we would use the *primal gap*:

$$\gamma_p'(\tilde{z}) = |\tilde{z} - z^\dagger|$$

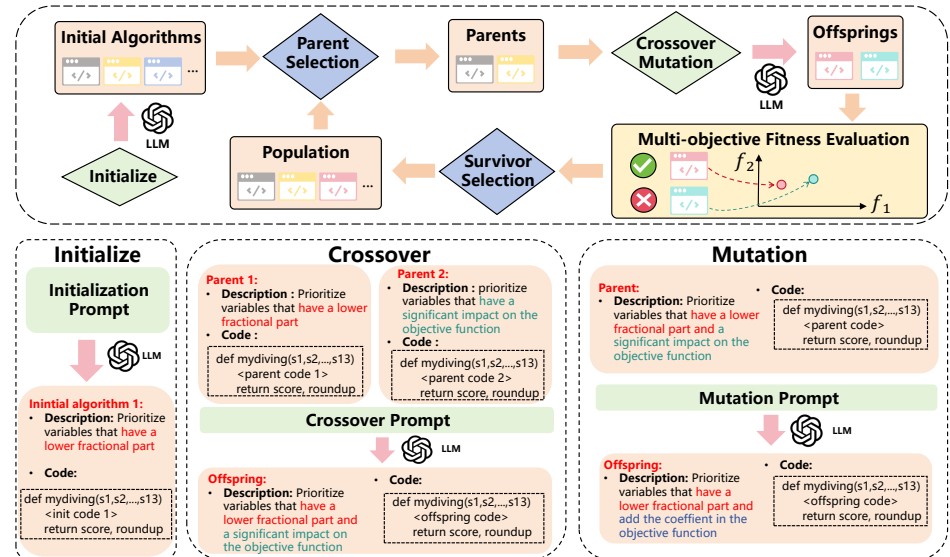

Figure 1: Illustration of the automatic algorithm design framework LLM4Solver. The **top** flowchart outlines the evolutionary process of the algorithms. LLM4Solver leverages the prior knowledge of LLM to generate new algorithm candidates in the initialization, crossover, and mutation steps. In the fitness evaluation step, LLM4Solver utilizes the solving performance of the candidates on one or *multiple* CO problems for single- or multi-objective evolution. The three parts at the **bottom** give examples of initialization, crossover, and mutation with LLM and prompt engineering.

## 3 METHODS

Modern CO solvers like SCIP (Gleixner et al., 2018) are highly complex, typically containing up to millions of lines of code. Thus, directly designing a whole CO solver end-to-end is challenging, as the search space grows exponentially with the algorithm complexity (Kuang et al., 2024b). Instead, in this paper, we mainly focus on the design of *diving heuristic*, which is widely recognized as one of the most critical primal heuristics in exact CO solvers to find high-quality solutions within a reasonably short time (Achterberg, 2007; Paulus & Krause, 2024).

Due to the large search space and lack of prior knowledge, previous manual and learning-based paradigms are inefficient for designing heuristics in CO solvers. Thus, we propose a novel framework (see Figure 1 and pseudo code 2)—large language model for exact combinatorial optimization solver (LLM4Solver)—to efficiently design high-quality and generic diving heuristics. The core idea of LLM4Solver is that, instead of defining complex symbols and searching in the space of symbolic trees (Poli et al., 2008; Petersen, 2019; Kuang et al., 2024a), we can *utilize the prior knowledge learned from large language models to conduct efficient searching directly at the program space*. We further extend this framework through multi-objective evolution to leverage the heterogeneous characteristics of different CO problems and design more generic heuristics.

Generally, as shown in Figure 1, LLM4Solver begins evolution with population initialization, iteratively optimizing the algorithms through the following steps: parent selection, crossover, mutation, fitness evaluation, and survivor selection. It leverages the prior knowledge of LLM to generate new algorithm candidates during initialization, crossover, and mutation (shown in the bottom part of Figure 1). During fitness evaluation, single-objective evolution uses one fitness function, $f_1$, while multi-objective evolution considers multiple fitness functions, $f_1, f_2, \ldots, f_n$, to assess performance across various CO problems and design a more generic algorithm.

Specifically, Section 3.1 first describes how to represent algorithms as individuals and how to evaluate the individuals in the evolution. Then, in Section 3.2 we introduce the idea of utilizing LLM to generate new individuals. After that, Section 3.3 describes the process of selecting the survivor individuals for the next generation of populations. Moreover, in Section 3.4 we introduce LLM4Solver with multi-objective evolution to simultaneously utilize information from different CO problems and design an algorithm with cross-benchmark generalization ability.

### 3.1 Individual Representation and Fitness Evaluation

As a population-based optimization strategy, each individual in the evolutionary process is represented by a diving score function $s$ and the description of its logic. The fitness of each individual is assessed based on its solving performance in specific instances. Throughout the evolutionary process, a population of $N$ individuals is maintained to facilitate optimization.

**Diving Score Function** The diving score function $s$ is the key decision-making component of the diving heuristic (See Appendix B). It determines the next diving variable and the rounding direction, which directly impacts the solving performance. For a diving score function $s$ with Python format, we employ a variable's 13 features as input and output *score* (float, the score of the variable) and *roundup* (bool, **True** if round the variable up, **False** for rounding down). These 13 features listed in Table 8 represent the union of all features used by the human-designed diving heuristics in SCIP. They are cheap to obtain, interpretable, and effectively describe the state of variables.

As previous methods based on neural networks(Paulus & Krause, 2024) and symbolic discovery(Kuang et al., 2024a) do not consider the description of algorithm logic, we treat both the diving score function and its logical description as an individual like (Liu et al., 2024). As shown in the bottom part of Figure 1, these logical descriptions provide a high-level idea for the corresponding algorithms, helping both the LLM and humans understand the algorithms. Through steps like crossover and mutation, LLM can combine and mutate these ideas to guide the generation of new algorithms.

**Fitness Evaluation** As primal heuristics aim to find better feasible solutions, we use the quality of solutions found by the diving heuristics as their fitness. Specifically, we embed $s$ in SCIP, turn off the other heuristics, dive into the root node, and the fitness is

$$f(s) = mean(\gamma_p(\tilde{z}_1^s), \gamma_p(\tilde{z}_2^s), ..., \gamma_p(\tilde{z}_{N_{ins}}^s)), \tag{1}$$

where $N_{ins}$ is the number of instances used for fitness evaluation, $\tilde{z}_k^s$ is the incumbent feasible solution found by $s$ in the $k$-th instance, $\gamma_p(\tilde{z}_k^s)$ is the relative primal gap of $s$ in the $k$-th instance. For an individual $s$, the smaller the value of $f(s)$, the better $s$ is.

### 3.2 Generating New Individuals with LLM

The text comprehension and logic generation abilities exhibited by LLMs closely align with the algorithm design. Therefore, we leverage the capabilities of LLM with prompt engineering to quickly generate new individuals in initialization, crossover and mutation steps.

**Prompt Engineering** As general LLM lacks sufficient information in specific domains, we need to provide more details of diving heuristics and specific instructions through prompt engineering. For the convenience of LLM's comprehension, we categorize the prompts into two groups: 1) the *background prompts* to provide the details about diving heuristics and 2) the *task-specific prompts* to instruct how to generate a new individual in initialization, crossover or mutation steps. Specifically, the background prompts consist of the introduction of MILP and diving heuristics and give the pseudo-code of diving for task-specific prompts. The task-specific prompts comprise the input features description, in-out format description and task-specific instruction. We give specific prompts in Appendix E.

**Initialization, Crossover and Mutation** In the initialization step, we need to leverage the capabilities of LLM to generate an individual from scratch. We directly use initialization-specific prompts and the LLM to generate a new individual. After running $N$ times, we can get the first population $P = \{s_1, s_2, ..., s_N\}$. In the crossover step, we need to recombine the advantages of the parent individuals to obtain a better offspring individual. Specifically, we provide $l$ parent algorithms and recombine them with crossover-specific prompts to generate $r$ offspring individuals. Then, we employ the LLM with mutation-specific prompts to mutate the offspring individuals generated by crossover, thereby exploring new individuals in the vicinity of the current one. We run the crossover and mutation for $N$ times in each generation and get $rN$ new individuals after that.

**Parent Selection** Before crossover, we need to select $l$ parent algorithms from the population. As parent selection needs to balance randomness and optimality, we adopt the *fitness proportional selection*(Zhou et al., 2019) to determine the probability of each individual $s_n$ being selected, i.e.

$$g(s_n) = \frac{\frac{1}{f(s_n)+eps}}{\sum_{k=1}^{N} \frac{1}{f(s_k)+eps}}, \quad n = 1, 2, ..., N, \tag{2}$$

where $eps$ is a small number to avoid division by $0$. It can be observed that the probability of better individuals being selected as parents is higher, which effectively balances randomness and optimality, allowing for exploration of a larger algorithm space.

### 3.3 ELITISM SURVIVOR SELECTION FOR POPULATION OPTIMIZATION

The Survivor Selection Policy plays a vital role in determining which individuals are kept and which are removed from the next generation. It is essential to ensure that the best individuals are preserved within the population. Specifically, we employ *elitism survivor selection*(Zhou et al., 2019) which always select the $N$ individuals with best fitness from the total $(r + 1)N$ ones after crossover and mutation. Elitism survivor selection can excellently ensure the optimality of individuals, which is helpful for efficiently finding high-quality diving heuristics.

### 3.4 EMPLOYING MULTI-OBJECTIVE EVOLUTION FOR CROSS-BENCHMARK GENERALIZATION

To design a unified algorithm with cross-benchmark generalization ability for the CO solver, we need to leverage the information of different benchmarks simultaneously. However, previous gradient-based work has only one optimization objective(Gasse et al., 2019; Kuang et al., 2024a;b), which is the algorithm's performance on a single benchmark, making it difficult to simultaneously utilize information from multiple benchmarks. To tackle this problem, we treat the performance of the algorithm on different benchmarks as separate objectives and utilize multi-objective evolution to harness information from different benchmarks. Specifically, the differences between single-objective and multi-objective are in the *fitness evaluation*, *parent selection*, and *survivor selection* steps.

**Fitness Evaluation of Multi-objective Evolution** We treat the fitness of the diving heuristics on different benchmarks as different objectives. Specifically, the $m$-th objective (fitness) is

$$f_m(s) = mean_m(\gamma_p(\tilde{z}_1^s), \gamma_p(\tilde{z}_2^s), ..., \gamma_p(\tilde{z}_{N_{ins\_m}}^s)), \quad m = 1, 2, ..., M, \qquad (3)$$

where $M$ is the number of objectives.

**Parent and Survivor Selection of Multi-objective Evolution** We use the *binary tournament selection*(Deb et al., 2002) for parent selection. Specifically, we randomly choose 2 individuals from the population and select the better one as a parent. Similarly, we use the *elitism survivor selection* for multi-objective evolution to select the best next generation. Since multi-objective optimization cannot simply compare two individuals using a single fitness function, we employ *Non-dominated Sorting* (See Appendix F) and *Crowding Distance Sorting* to perform the necessary comparisons.

## 4 EXPERIMENTS

We evaluate LLM4Solver through extensive experiments and benchmarks[1]. These experiments aim to 1) show that LLM4Solver with single-objective evolution algorithm (SOEA) significantly outperforms current human-designed and learning-based SOTA methods in solution quality and solving efficiency; 2) illustrate that LLM4Solver with multi-objective evolution algorithm (MOEA) designs heuristics with high cross-benchmark generalization ability; 3) show the interpretability of LLM4Solver; 4) conduct ablation studies to highlight the effectiveness of evolution search.

### 4.1 EXPERIMENTAL SETTINGS

**Baselines** There are 8 baselines corresponding to solution quality, including 7 human-designed diving heuristics (i.e. coefficient, fractional, linesearch, pseudocost, distributional, vectorlength and farkas(Witzig & Gleixner, 2021) diving) in the open-source solver SCIP(Achterberg, 2007) and a learning-based GNN diving method L2DIVE(Paulus & Krause, 2024). For solving efficiency, we compare our method with the default and tuned SCIP and L2DIVE. Specifically, we do not change any parameters for the default SCIP, but we tune two important parameters (i.e. *freq* and *freqofs*) for tuned SCIP to get better performance. See Appendix D.1 for more details.

**Benchmarks** The same as previous work(Paulus & Krause, 2024), we employ four standard and two real-world benchmarks to compare the solution quality and solving efficiency. The four standard

---

[1]We report the learned heuristics in Appendix G and will release our training code once the paper is accepted.

Table 1: The average relative primal gap with standard error of different diving heuristics. The results compare LLM4Solver with SOEA to seven human-designed and one learning-based baselines to illustrate the superior quality of solutions found by the designed diving heuristics.

| Methods | Setcover | Cauctions | Facilities | Indset |
|---|---|---|---|---|
| LLM4Solver with SOEA | **3.36 (0.25)** | **1.83 (0.16)** | **0.65 (0.03)** | **0.84 (0.07)** |
| Best Human-designed | *6.99 (0.38)* | *3.00 (0.21)* | *2.17 (0.09)* | *4.91 (0.45)* |
| coefficient | 232.47 (3.47) | 8.10 (0.34) | 5.61 (0.18) | 15.49 (0.50) |
| distributional | 231.54 (3.45) | 9.47 (0.40) | 3.11 (0.11) | 11.30 (0.25) |
| farkas | *6.99 (0.38)* | 5.67 (0.29) | 2.19 (0.09) | – |
| fractional | 232.43 (3.47) | 7.63 (0.31) | 5.61 (0.18) | 14.55 (0.40) |
| linesearch | 232.43 (3.47) | 3.58 (0.26) | 6.78 (0.31) | 10.51 (0.51) |
| pseudocost | 18.62 (1.47) | *3.00 (0.21)* | *2.17 (0.09)* | 9.82 (0.49) |
| vectorlength | 232.43 (3.47) | 61.67 (0.55) | 6.78 (0.31) | *4.91 (0.45)* |
| L2DIVE[2] | 3.58 | 2.60 | 0.71 | 1.37 |

ones include set covering (Setcover), combinatorial auctions (Cauctions), capacitated facility location (Facilities), and maximum independent sets (Indset), and the two real-world ones include server load balancing in distributed computing (LoadBalance)(Gasse et al., 2022) and neural network verification (NNVerify)(Nair et al., 2020). We utilize the heterogeneous benchmark MIPLIB2017 containing 20 instances(Gleixner et al., 2021) to further demonstrate the cross-benchmark generation ability. These 20 instances (See Table 10) ensure that at least one diving heuristic can find a feasible solution, facilitating the comparison of different diving heuristics. We report the size of benchmarks and the hyperparameters for generating them in Appendix D.2.

**Implementation Details** (1) For the solution quality, Diving is solely implemented in the root node of each instance, with branching, cutting planes, and other primal heuristics disabled, emphasizing the quality of feasible solutions found by diving heuristics. For fitness evaluation in the evolution, we generate 50 instances each for Setcover, Cauctions, and Indset, and 10 instances for Facilities. We validate and test the discovered diving heuristics on 100 instances each. (2) For the solving efficiency, we embed the discovered diving heuristics into the SCIP. We use the LoadBalance dataset(Gasse et al., 2019) with 100 instances for validation and testing respectively. These instances cannot be solved within 3600 seconds, so we set a limit time $T_{limit} = 900$ seconds and measure its primal-dual integral $PD(T_{limit})$ as the solving performance. We use the NNVerify dataset with 50 instances for validation and 523 for testing. We set the maximal solving time limit $T_{limit} = 3600$ seconds and measure the solving time $T$. We use solution quality as the evaluation criterion and then utilize the primal-dual integral or solving time for validation and testing, selecting the diving heuristic that demonstrates the best performance in solving efficiency. (3) For LLM4Solver with multi-objective evolution algorithm (MOEA), we set the primal gap on four benchmarks (Setcover, Cauctions, Facilities, and Indset) as four objectives. Then, we chose the diving heuristic in the Pareto front with the highest average improvement ratio compared to the human-designed heuristic on four benchmarks as the output. We employ the designed diving heuristic to the MIPLIB instances to show the cross-benchmark generalization ability. We report the mean of the primal gap and the wins for comparison.

We use the GPT-3.5-turbo as the pre-trained LLM. We run all the experiments with 3 rand seeds on Intel(R) Xeon(R) CPU E5-2667 v4 @ 3.20GHz and NVIDIA GeForce RTX 2080 Ti.

## 4.2 RESULTS

**Solution Quality** We compare LLM4Solver to other baselines in Table 1. Results show that LLM4Solver with single-objective evolution algorithm (SOEA) finds better feasible solutions than *all* human-designed and learning-based SOTA diving heuristics on four different problem classes. The results show that combining the prior knowledge of LLMs and evolutionary search is effective for designing new algorithms of CO solvers. We further expose the results of harder instances in Appendix D.3 to show the scalability of LLM4Solver.

---

[2]Since the code for L2DIVE is currently not open-source and specific hyperparameters are not available, we officially report the performance of L2DIVE based on its ratio to the best human-designed heuristic as presented in the original article (Paulus & Krause, 2024). For a fair comparison, we use the same benchmarks and observe consistent results for the human-designed baselines.

Table 2: Compare SCIP with LLM4Solver to default and tuned SCIP to illustrate that it can improve the quality of solutions while leveraging better solutions to enhance the solving efficiency. LLM4Solver improves the primal-dual integral by 38% (15%) on LoadBalance and reduces the solving time by 31% (20%) on NNVerify over the default (tuned) settings of SCIP.

| | LoadBalance | | NNVerify | |
|---|---|---|---|---|
| | Primal-dual Integral | Wins | Solving Time | Wins |
| Default SCIP | 7340.7 (58.1) | 0 (0.0) | 76.9 (4.08) | 44 (9.2) |
| Tuned SCIP | 5445.7 (100.8) | 1 (0.5) | 65.8 (1.09) | 103 (10.1) |
| SCIP with LLM4Solver | **4543.0 (53.1)** | **99 (0.5)** | **52.9 (1.49)** | **376 (14.6)** |

Table 3: The average relative primal gap of LLM4Solver with MOEA, LLM4Solver with SOEA, and human-designed diving heuristics. The results show that using multi-objective evolution can leverage the characteristics of different CO problems to achieve cross-benchmark generalization ability.

| Methods | Setcover | Cauctions | Facilities | Indset |
|---|---|---|---|---|
| LLM4Solver with MOEA | 4.01 (0.32) | 2.49 (0.19) | 1.30 (0.08) | 1.28 (0.11) |
| LLM4Solver trained on Setcover | **3.36 (0.25)** | 2.77 (0.21) | 3.33 (0.18) | 9.75 (0.49) |
| LLM4Solver trained on Cauctions | 5.30 (0.36) | **1.83 (0.16)** | 3.28 (0.20) | 14.87 (0.48) |
| LLM4Solver trained on Facilities | 6.86 (0.54) | 11.37 (1.21) | **0.65 (0.03)** | 5.93 (0.35) |
| LLM4Solver trained on Indset | 70.96 (1.50) | 11.52 (0.52) | 3.18 (0.16) | **0.84 (0.07)** |
| Best Human-designed | 6.99 (0.38) | 3.00 (0.21) | 2.17 (0.09) | 4.91 (0.45) |
| farkas | 6.99 (0.38) | 5.67 (0.29) | 2.19 (0.09) | – |
| pseudocost | 18.62 (1.47) | 3.00 (0.21) | 2.17 (0.09) | 9.82 (0.49) |
| vectorlength | 232.43 (3.47) | 61.67 (0.55) | 6.78 (0.31) | 4.91 (0.45) |

**Efficient Searching** We show the convergence process of LLM4Solver with SOEA on Setcover dataset in Figure 2. It illustrates that LLM4Solver can design an algorithm better than the best human-designed ones in the *first* generation and better than the SOTA learning-based method L2DIVE in the *fourth* generation. The convergence time with 10 iterations is 3503.5±73.4s for Setcover, and 1835.4±56.6s, 4568.2±116.3s, 1213.6±48.9s for Cauctions, Facilities, and Indset respectively. The result shows that LLM4Solver is *efficient* for designing high-quality diving heuristics.

**Solving Efficiency** We compare LLM4Solver to default and tuned SCIP in Table 2. Results show that LLM4Solver improves the solution quality while leveraging better solutions to enhance the solving efficiency. LLM4Solver improves the primal-dual integral by 38% (15%) on LoadBalance and reduces the solving time by 31% (20%) on NNVerify over the default (tuned) settings of SCIP comparing that L2DIVE improves 35% (7%) on LoadBalance and 29% (20%) on NNVerify.

**Generalization Ability of LLM4Solver with MOEA** In Table 3, we compare the performance of individual algorithms across multiple CO problems. For example, the row labeled "LLM4Solver trained on Setcover" represents the performance of the algorithm trained on the Setcover problem across four different problems. In Figure 3, we use a radar plot to visually compare the performance of MOEA, SOEA, and human-designed heuristics across different CO problems. The results in Table 3 and Figure 3 show that 1) although the diving heuristics designed by LLM4Solver with SOEA perform well on their corresponding benchmarks, they struggle to generalize ef-

Table 4: Compare LLM4Solver to human-designed diving heuristics to illustrate high generalization ability on heterogeneous MIPLIB of 20 instances. For "Wins", "8/18" means the heuristic can find feasible solutions on "18" instances and get the best solutions on "8".

| Heuristics | Average Primal Gap | Wins |
|---|---|---|
| coefficient | 1510 | 0/18 |
| distributional | 4826 | 1/15 |
| farkas | 1180 | 3/11 |
| fractional | 1268 | 1/14 |
| linesearch | 1589 | 1/16 |
| pseudocost | 1242 | 4/15 |
| vectorlength | 4803 | 2/17 |
| LLM4Solver | **844** | **8/18** |

fectively to other benchmarks (e.g., the algorithms evolved using Indset perform poorly on Setcover); 2) LLM4Solver with MOEA consistently outperforms the best human-designed heuristics across all datasets and demonstrates better cross-benchmark generalization ability compared to single-objective evolution. Moreover, we measure the performance of the diving heuristic designed by MOEA on 20 MIPLIB instances, results in Table 4 show that it can find better solutions even on heterogeneous

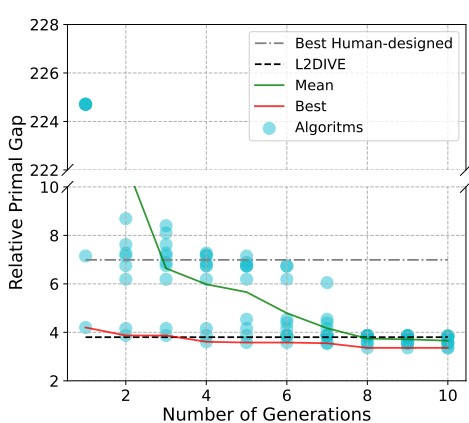 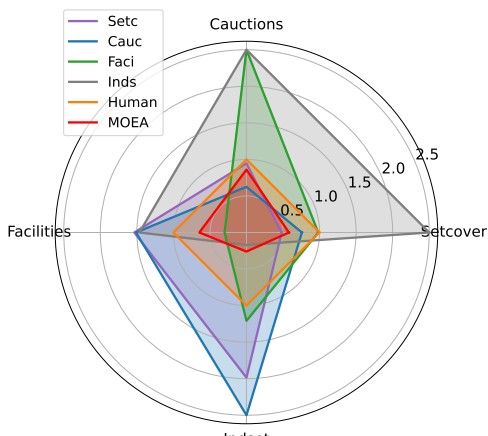

Figure 2: The convergence curve of LLM4Solver with SOEA on the Setcover problem, where each point represents an algorithm during the evolution. The x-axis represents the iterations, the y-axis indicates the solution quality. The red and green lines represent the best and mean relative primal gap per generation. The black and grey dotted lines represent L2DIVE and the best human-designed diving heuristic.

Figure 3: The radar plot comparison on the performance of MOEA, SOEA, and human-designed heuristics with different CO problems. The radius represents the ratio of each algorithm's relative primal gap compared to the human-designed one (we set the maximum radius to 2.5 for visualization). Therefore, a smaller radius or enclosed area indicates better performance. In this context, "Setc" refers to "LLM4Solver trained on Setcover," and "MOEA" refers to "LLM4Solver with MOEA".

and unseen instances. This result demonstrates that LLM4Solver with MOEA can simultaneously utilize characteristics from different CO problems to design a unified algorithm with cross-benchmark generalization ability, which is beneficial to improving the solvers' built-in capabilities.

**Interpretability** We report the diving heuristic designed by LLM4Solver with MOEA in Figure 4. There are three key features that enable LLM4Solver to achieve both interpretability and high performance. (1) Leveraging LLMs' text and code generation capabilities, LLM4Solver directly generates code and provides comments for the code. This offers greater interpretability compared to black-box neural networks (Paulus & Krause, 2024) and purely numerical symbolic methods (Kuang et al., 2024a). (2) Most of the intuition in LLM4Solver aligns with human reasoning. For example, the condition to determine the rounding direction is " candsfrac > 0.5 ". Since "candsfrac" represents the fractional part of an integer variable, the intuitive approach is to round up if it is closer to 1 and down if closer to 0. (3) LLM4Solver can fine-tune parameters or specific computational methods to achieve higher performance for particular problems. For example, after multiple rounds of evolution, it ultimately selected a penalty value

```python
def myheurdiving(mayrounddown, mayroundup, candsfrac, candsol,
nlocksdown, nlocksup, obj, objnorm, pscostdown,  pscostup, rootsolval,
nNonz, isBinary):
    # Initialization
    score = 0.0
    roundup = False
    # Penalize limited rounding options to encourage exploration
    score -= 0.2 if mayrounddown or mayroundup else 0.0
    # Prioritize variables with high fractional values and low pseudo costs
    score += candsfrac
    score += min(1 / (1 + (pscostdown + pscostup)), 1)
    # Consider the impact of objective function value and Euclidean norm
    score += (obj / max(1, objnorm)) * (1 - candsfrac) / (nNonz + 1)
    # Incorporate historical solution values
    score += rootsolval * 0.3
    # Penalize excessive sparsity
    if nNonz < 5:
        score -= 0.2
    # Adjust based on the number of locks for rounding down/up of a
special type
    score -= min(0.2 * (nlocksdown + nlocksup), 0.4)
    # Differentiate based on the binary nature of the variable
    score *= 1.5 if isBinary else 1.0
    # Determine rounding direction based on the score
    if candsfrac > 0.5:
        roundup = True
    return score, roundup
```

Figure 4: The code designed by LLM4Solver

of "0.2" for the "penalize limited rounding options" strategy. Similarly, for "prioritize low pseudo costs," it uses the formula "min(1/(1+(pscostdown+pscostup)), 1)" instead of directly applying "-(pscostdown+pscostup)". These algorithms not only improve the solving performance but also help

experts obtain **insights** into solving patterns. The insights potentially play a significant role in the design of the next generation of solvers.

### 4.3 ABLATION STUDY

We conduct ablation studies to provide more evidence of the contribution of different parts in LLM4Solver. First, we compare the contribution of different parts in the evolutionary process. For "LLM (No Evolution)", we disable all evolutionary processes and solely use the LLM to generate 100 algorithm candidates. Additionally, we compare LLM4Solver where

Table 5: A comparison of different parts in evolution. The average relative primal gap with standard error.

| Methods | Setcover | Cauctions |
| --- | --- | --- |
| LLM (No Evolution) | 3.84 (0.31) | 2.84 (0.22) |
| LLM4Solver (No Crossover) | 3.67 (0.28) | 2.77 (0.21) |
| LLM4Solver (No Mutation) | 3.55 (0.27) | 2.34 (0.19) |
| LLM4Solver | 3.36 (0.25) | 1.83 (0.16) |

either crossover or mutation was excluded individually. The results in Table 5 indicate that crossover is more crucial for the outcomes. Without crossover, the solution quality decreases by 10%-28%, while not using any evolutionary process results in a 14%-55% reduction in solution quality.

Second, we compare LLM4Solver with different LLMs (GPT-4, GPT-3.5-turbo-16k, Claude-3.5-sonnet). The results in Table 6 show that the performance of LLM4Solver is not entirely dependent on the reasoning capability of the LLMs and all four LLMs can achieve high performance. This further indicates that the evolutionary search framework in LLM4Solver helps compensate for the differences in reasoning capabilities among the various LLMs.

Table 6: A comparison of different LLMs. The average relative primal gap with standard error.

| LLM | Setcover | Cauctions |
| --- | --- | --- |
| GPT-4 | 3.43 (0.28) | 2.29 (0.18) |
| GPT-3.5-turbo-16k | 3.48 (0.28) | 2.16 (0.17) |
| Claude-3.5-sonnet | 3.41 (0.27) | 2.10 (0.18) |
| GPT-3.5-turbo | 3.36 (0.25) | 1.83 (0.16) |

Finally, we compare the impact of key hyperparameters during the evolutionary process on the final results, including the number of generations ($N_g$) and population size ($N$). The results in Table 7 indicate that once convergence is achieved, increasing the number of generations or population size does not significantly improve the final results.

Table 7: A comparison of different hyperparameters. The average relative primal gap with standard error.

| LLM | Setcover | Cauctions |
| --- | --- | --- |
| $N_g = 10, N = 10$ | 3.36 (0.25) | 1.83 (0.16) |
| $N_g = 20, N = 10$ | 3.34 (0.23) | 1.91 (0.17) |
| $N_g = 10, N = 20$ | 3.31 (0.23) | 1.80 (0.21) |

## 5 CONCLUSIONS

In this paper, we propose a novel LLM-based automatic algorithm design framework for combinatorial optimization solvers to efficiently design high-quality and generic diving heuristics. To leverage the heterogeneous characteristics of different CO problems, we extend this framework through *multi-objective evolution*. Extensive experiments show that LLM4Solver significantly outperforms all the SOTA human-designed and learning-based (on GPU) methods in terms of solution quality, solving efficiency, and cross-benchmark generalization ability. Furthermore, the appealing features of LLM4Solver include high performance, efficient searching, and interpretability of the designed algorithms. The results show an encouraging step towards efficient automatic algorithm design on modern exact CO solvers via large language models. Applying LLM4Solver to more components in modern CO solvers like branching (Gasse et al., 2019; Kuang et al., 2024a), presolve (Kuang et al., 2023; Achterberg, 2007), and cut generation (Huang et al., 2022; Wang et al., 2023) are exciting avenues for further work. Moreover, the automated algorithm design framework based on multi-objective evolution can be extended to more complex problems like multi-objective CO problems(Chen et al., 2024; Lust & Teghem, 2010) and electronic design automation(Wang et al., 2024). Finally, LLM4Solver shows the potential to efficiently design high-quality and generic algorithms for the next generation of solvers, thereby enhancing their built-in capabilities.

## 6 REPRODUCIBILITY STATEMENT

We do all experiments on the open-source CO solver SCIP Optimization Suite 9.0 (Bolusani et al., 2024). For benchmarks, we provide the information of four standard and two real-world benchmarks in 4.1 and D.2. We give the name of used MIPLIB instances in 10. For baselines, we give all the information about human-designed diving heuristics in B and source code is in SCIP(Bolusani et al., 2024). For our methods, we give LLM4Solver's pseudo-code in 2, hyper-parameters in C, prompts in E and the designed diving heuristics in G. We will **release all the codes** for training and evaluation once the paper is accepted.

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

## A    RELATED WORKS

Existing CO solvers consist of branching, cut selection, primal heuristics, and other essential modules (Achterberg, 2007). The application of Machine Learning (ML) leverages its inherent capabilities to learn data distributions and replace one or more modules within the solver to enhance solving efficiency. The research on ML for CO solvers can be roughly categorized into two classes. One class aims to substitute modules with deep neural networks, for example, Gasse et al. (2019) propose a bipartite graph presentation of CO problems and replace the branching module with the graph neural network (GNN). Nair et al. (2020) construct a *Neural Solver* to generate a high-quality assignment and optimality gap. Han et al. (2023) leverage the predict-and-search framework to generate better assignment. Wang et al. (2023) develop a hierarchical sequence/set model to learn cut selection policies. While previous works achieved significant results, complex models face disadvantages including high requirement of training samples, low interpretability, and great deployment difficulty (Kuang et al., 2024a). To tackle these problems, the other class aims to discover symbolic algorithms in a data-driven methodology. Kuang et al. (2024a) leverage deep symbolic regression to learn branching policies in expressions, outperforming previous neural network methods on purely CPU-based devices. Symbolic learning-based methods also have broad prospects for scientific discovery. Chen et al. (2023) propose program search techniques and discovered a more efficient optimization algorithm *Lion*. Mankowitz et al. (2023) train an agent *AlphaDev* to learn sorting algorithms and achieve better performance than human benchmarks. Although the discovery process runs automatically, training these works still requires developing a novel algorithm from scratch, which is inefficient with limited prior knowledge.

As the performance of Large Language Models grows rapidly (Naveed et al., 2023), researchers attempt to combine the prior knowledge of Large Language Models (LLMs) with algorithm design. Yang et al. (2023) propose an approach that iteratively optimizes prompts and solutions to problems by LLMs. Xiao & Wang (2023) leverage LLMs to assist the design of the robotic modules and attain the utility-optimal A* algorithm. Despite the excellent capabilities of LLMs, it is still tough to design algorithms for complex problems with solely LLMs and prompt engineering. Thus, recent works consider evolutionary search methods and obtain more competitive algorithms through the continuous iterative evolution of algorithms generated by LLM. Romera-Paredes et al. (2024) combine LLM with the island-based evolutionary method, and discover new algorithms for classic mathematical problems like cap set and online bin packing, revealing the great potential of this combination. Liu et al. (2023) introduce a novel approach *AEL* and discover heuristics with excellent generalization performance on the traveling salesman problem. Ye et al. (2024) leverage LLM to revise heuristics for online bin packing, traveling salesman problems, and electronic design automation. Sun et al. (2024) create a multi-agent-based framework to improve the heuristics of SAT problems. These works have demonstrated impressive results in scenarios like mathematical discovery, and designing heuristics for classical CO and SAT problems. However, while online bin packing and traveling salesman problems are highly representative and important CO problems, when modeled and solved using general MILP formulations in CO solvers, it is critical to investigate heuristics to find feasible solutions in more general MILP problems. Especially in the domain of exact CO solver, various cross-distribution problems, such as Set covering (Balas & Ho, 1980), Combinatorial auction (Leyton-Brown et al., 2000), Capacitated facility location (Cornuéjols et al., 1991), Maximum independent set (Bergman et al., 2015), etc., are modeled by a unified format for solving. Hence, it is essential to design a general algorithm that can cater to diverse problem types for exact CO solvers.

## B    DIVING HEURISTICS AND INPUT FEATURES

Every diving heuristic shares the same generic framework 1 and the only difference is the score function $s$ to decide the rounding variables and direction.

There are some human-designed diving heuristics used in SCIP.

**Coefficient Diving** selects the variable that minimizes the number of up-lock or down-lock constraints and bounds it in the up or down direction. We say a constraint $C$ is up-lock (down-lock) on $x_j$ if $C(\hat{x}) = 1, C(\tilde{x}) = 0$, where $\hat{x}_k = \tilde{x}_k$ for all $k \neq j$ and $\hat{x}_j > \tilde{x}_j$ ($\hat{x}_j < \tilde{x}_j$).

**Distributional Diving** selects a variable according to the solution density obtained by fixing its value, which is proposed in Pryor & Chinneck (2011).

Table 8: The features of the variables used for diving score functions

| Features | Name | Description |
|---|---|---|
| s1 | mayrounddown | bool, indicate whether it is possible to round variable down and stay feasible. |
| s2 | mayroundup | bool, indicate whether it is possible to round variable up and stay feasible. |
| s3 | candsfrac | float, fractional part of solution value of variable. |
| s4 | candsol | float, solution value of the variable in LP relaxation solution. |
| s5 | nlocksdown | int, the number of locks for rounding down of a special type. |
| s6 | nlocksup | int, the number of locks for rounding up of a special type. |
| s7 | obj | float, objective function value of variable. |
| s8 | objnorm | float, the Euclidean norm of the objective function vector. |
| s9,s10 | pscostdown/up | float, the variable's pseudo cost value for the given change of the variable's LP value. |
| s11 | rootsolval | float, the solution of the variable in the last root node's relaxation. |
| s12 | nNonz | int, number of nonzero entries in the column vector. |
| s13 | isBinary | bool, TRUE if the variable is of binary type. |

**Farkas Diving** (Witzig & Gleixner, 2021) comes from the Farkas' lemma. It bounds a variable along the direction in which the objective is improved and selects the variable with the largest lifting objective value.

**Fractionality Diving** selects the variable with minimal fractionality $\min\{\lceil x_j\rceil - x_j, x_j - \lfloor x_j\rfloor\}$ and rounds it to the nearest integer.

**Line Search Diving** reinforces the solution update direction from the root node. It selects the first variable that reaches an integer on the ray from the LP solution at the root to the LP solution at the current node. Algebraically spoken, it rounds the variable having the minimal distance ratio $\frac{\lceil x_j\rceil - x_j}{x_j - (x_{root})_j}$ for $x_j > (x_{root})_j$ and $\frac{x_j - \lfloor x_j\rfloor}{(x_{root})_j - x_j}$ for $x_j < (x_{root})_j$.

**Pseudocost Diving** selects a variable based on its pseudocost collected during the search process. The pseudocosts provide an estimation for each integer variable, indicating the increase in the objective value of the LP problem per unit change in that variable.

**Vector Length Diving** is custom-made for set covering and applied to general MIPs. It selects the variable with the smallest ratio of the objective cost to the constraints covered by fixing it to 1.

There are some other diving heuristics in SCIP, including *adaptive* diving, *guided* diving and *conflict* diving. We do not compare these diving heuristics as baselines because they are ineffective (*conflict*), require at least one feasible solution (*guided*), or choose from other heuristics (*adaptive*).

In our work, we take the variable's features used in these human-designed diving heuristics as input for our diving score function. See Table 8 for details of the total 13 features.

---

**Algorithm 1** Generic Diving Heuristic

**Input:** MILP with relaxation constraints $P^*$, LP solution $x^*$, maximum depth $d_{max}$
**Output:** If available, a set of feasible solutions $X$
**Require:** a scoring function $s$ for selecting tighten variables and corresponding round direction

1: Initial depth $d \leftarrow 1$, $\mathcal{C} := \{j \in \mathcal{I} | x_j^* \notin \mathbb{Z}\}$
2: **while** $d \leq d_{max}$ **do**
3:   $j = \arg\max_{i\in\mathcal{C}} s(x_i)$
4:   $l_j \leftarrow \lceil x_j^*\rceil$ if roundup else $u_j \leftarrow \lfloor x_j^*\rfloor$
5:   $P^* \leftarrow P^* \cap \{l_j \leq x_j \leq u_j\}$
6:   **if** $P^*$ is infeasible **then** break
7:   $x^* = \arg\min_x\{\mathbf{c}^\top\mathbf{x} | \mathbf{x} \in P^*\}$
8:   **if** $x^*$ is roundable **then**
9:    $X \leftarrow X \cup round(x^*)$
10:   **end if**
11:   $d \leftarrow d + 1$
12:   update candidate variable index set $\mathcal{C}$
13: **end while**

---

## C    PSEUDO-CODE OF LLM4SOLVER AND HYPERPARAMETERS

We provide the pseudo-code of LLM4Solver in Algorithm 2. Firstly, during the initialization, we utilize an LLM to generate the initial population $P = \{s_1, s_2, \ldots, s_N\}$. Subsequently, through the use of parent selection, crossover, mutation, fitness evaluation, and survivor selection, we iteratively evolve the algorithm's population. After $N_g$ generations of iteration, we can get one high-performance diving score function $s$ for the exact combinatorial optimization solver. We list the hyperparameters as follows: $N_g(SOEA) = 10, N_g(MOEA) = 20, N_g(SOEA) = 10, N_g(MOEA) = 16, l = 2, r = 1, eps = 10^{-8}$.

---

**Algorithm 2** Large Language Models for Exact Combinatorial Optimization Solvers (LLM4Solver)

---

**Input:**  A given LLM; The number of generations: $N_g$; Population size $N$; The number of parents $l$; The number of new individuals $r$ generated by crossover; The number of objectives M.
**Output:**  Best diving score function $s^*$

 1: **for** $j = 1, 2, ..., N$ **do**
 2:     **Initialization:**  Creat new diving score function $s_j$ as individuals with given LLM;
 3:     **Fitness Evaluation:**  Evaluate its fitness $f_1(s_j), ..., f_M(s_j)$ with instances;
 4: **end for**
 5: Initial population $P = \{s_1, s_2, ..., s_N\}$
 6: **for** $i = 1, 2, ..., N_g$ **do**
 7:     **for** $j = 1, 2, ..., N$ **do**
 8:         **Parent Selection:**  Select the parent individuals $p_j = \{s_1, s_2, ..., s_l\}$
 9:         **Crossover:**  Create new individuals $o_j = \{s_1, s_2, ..., s_r\}$ with LLMs, crossover
10:         prompts and $p_j$
11:         **for** $k = 1, 2, .., r$ **do**
12:             **Mutation:**  Mutate $s_k$ with LLMs and mutation prompts
13:             **Fitness Evaluation:**  Evaluate its fitness $f_1(s_k), ..., f_M(s_k)$ with given instances;
14:         **end for**
15:     **end for**
16:     **Survivor Selection:**  Select the best $N$ individuals from $P \cup \{o_1, o_2, ..., o_N\}$ to generate
17:     the next population $P$
18: **end for**
19: Select the best $s^*$ from the latest population by validation as output.

---

## D    MORE EXPERIMENT DETAILS

### D.1    TUNING SCIP PARAMETERS FOR DIVING

There are two most important parameters *frep* and *freqofs* that control the stages where different diving heuristics take effect. Hence for baseline *Tuned SCIP*, we sample the configurations by varying these parameters to associate diverse heuristics and improve the performance. Learning from Paulus & Krause (2024), we define the sample distribution of *frep* by setting $freq = -1$ (no diving), $freq = 0.5 \times freq_{default}$ (double frequency), $freq = freq_{default}$ (default frequency), $freq = 2 \times freq_{default}$ (half frequency) with equal probability and the distribution of *freqofs* by leaving the $freqofs = 0$ and $freqofs = freqofs_{default}$ with equal probability for each diving heuristic. Under the same resource load and validation instances, we select the configuration with the lowest primal-dual integral and solving time for the baseline. Although tuning solver parameters may enhance our LLM4Solver, we keep the default settings to give the direct improvement reflection.

### D.2    BENCHMARK DETAILS

We follow the benchmark generation process in Gasse et al. (2019) for the problems including set covering (Setcover), capacitated facility (Facilities), combinatorial auction (Cauctions), and maximum independent set (Indset). We set 2 levels (i.e. easy, and hard) of difficulty according to the problem scales. We list the generation hyperparameters and algorithms in Table 9.

Table 9: Instance generation algorithms and the detailed hyperparameters.

| Benchmark | Algorithms | Hyperparameters | |
|---|---|---|---|
| Setcover | Balas & Ho (1980) | Easy: | 500 rows 1000 columns |
| | | Hard: | 2000 rows 1000 columns |
| Cauctions | Leyton-Brown et al. (2000) | Easy: | 100 items for 500 bids |
| | | Hard: | 300 items 1500 bids |
| Facilities | Cornuéjols et al. (1991) | Easy: | 100 facilities with 100 customers |
| | | Hard: | 100 facilities with 400 customers |
| Indset | Bergman et al. (2015) | Easy: | 500 nodes with affinity 4 |
| | | Hard: | 1500 nodes with affinity 4 |

Table 10: Used MIPLIB instance names

| | | | |
|---|---|---|---|
| air05 | beasleyC3 | binkar10_1 | cod105 |
| dano3_3 | eil33-2 | hypothyroid-k1 | istanbul-no-cutoff |
| markshare_4_0 | mas76 | mc11 | mik-250-20-75-4 |
| n5-3 | neos-860300 | neos-957323 | neos-1445765 |
| nw04 | piperout-27 | pk1 | seymour1 |

For load balancing in distributed computing (LoadBalance), we get the dataset the same as Gasse et al. (2022). We don't use the training set and we only use 100 instances for validation and testing respectively. For neural network verification (NNVerify)(Nair et al., 2020), we select 50 instances for validation and 523 for testing by excluding the unsolved, trivial and numerically unstable instances.

For MIPLIB instances, we choose the easy instances from MIPLIB2017 (Gleixner et al., 2021) that can be solved within 100s and at least one diving heuristic can find a feasible solution. The specific names are listed in Table 10.

### D.3 SCALE TO HARD INSTANCES

We compare the diving heuristics generated by LLM4Solver on easy instances and scale them to hard ones. Results in Table 11 show that LLM4Solver with SOEA still outperforms all human-designed heuristics on all four problems. It illustrates that LLM4Solver has a high scalability to hard instances. However, as mentioned in Table 3 of the main text, although LLM4Solver with SOEA can learn the characteristics of a single problem and generalize to harder instances of the same problem type, it is unable to learn features across different problems. Therefore, it lacks cross-benchmark generalization ability.

Table 11: LLM4Solver still outperforms all seven human-designed diving heuristics on *hard* test instances even trained on *easy* instances.

| Type | Heuristics | Setcover | Cauctions | Facilities | Indset |
|---|---|---|---|---|---|
| LLM4Solver: | gpt35 | 5.03 (0.26) | 2.29 (0.11) | 1.39 (0.24) | 0.80 (0.04) |
| | gpt35-16k | 6.34 (0.32) | **1.55 (0.09)** | 0.61 (0.02) | 0.79 (0.04) |
| | gpt4 | 5.66 (0.30) | 2.00 (0.10) | **0.42 (0.04)** | 0.79 (0.04) |
| | claude3 | **4.90 (0.26)** | 1.60 (0.07) | 0.43 (0.04) | **0.77 (0.04)** |
| Human-designed: | best human-designed | *9.49 (0.35)* | *3.17 (0.12)* | *3.04 (0.16)* | *4.00 (0.51)* |
| | coefficient | 332.39 (3.31) | 8.23 (0.20) | 28.15 (1.22) | 16.36 (0.30) |
| | distributional | 332.36 (3.31) | 12.83 (0.32) | 9.92 (0.54) | 10.93 (0.16) |
| | farkas | *9.49 (0.35)* | 5.82 (0.13) | 7.21 (0.48) | – |
| | fractional | 332.39 (3.31) | 8.84 (0.20) | 28.57 (1.25) | 15.23 (0.24) |
| | linesearch | 278.48 (1.93) | 3.78 (0.14) | 30.01 (1.18) | 14.16 (0.28) |
| | pseudocost | 25.59 (1.33) | *3.17 (0.12)* | *3.04 (0.16)* | 13.11 (0.26) |
| | vectorlength | 253.60 (2.38) | 60.85 (0.34) | 30.01 (1.18) | *4.00 (0.51)* |

Table 12: The relative primal gap between "average improvement ratio as single objective" and "LLM4Solver with MOEA + average improvement ratio".

| Methods | Setcover | Cauctions | Facilities | Indset |
|---|---|---|---|---|
| MOEA + Average Improvement Ratio | 4.01 (0.32) | 2.49 (0.19) | 1.30 (0.08) | 1.28 (0.11) |
| Average Improvement Ratio as Single Objective | 4.57 (0.42) | 2.54 (0.23) | 1.37 (0.12) | 1.38 (0.15) |

### D.4 TAKE AVERAGE IMPROVEMENT RATIO AS SINGLE OBJECTIVE

When selecting algorithms with cross-benchmark generalization ability from the Pareto Front obtained through multi-objective evolution, we used the average improvement ratio of the algorithms on four CO problems as a posterior selection criterion. This is defined as:

$$AIR(s) = \sum_{m=1}^{M} \frac{1}{M} \frac{f_m(s)}{h_m}, m = 1, 2, ..., M \tag{4}$$

where $AIR(s)$ is the average improvement rario of $s$, $M$ is the number of CO problems (objectives), $f_m(s)$ and $h_m$ are the relative primal gap of $s$ and best human-designed algorithm respectively on the $m$-th CO problem.

We use the average improvement ratio as a single objective for LLM4Solver with SOEA, and the results in Table 12 show that LLM4Solver with MOEA outperforms the approach that relies solely on the average improvement ratio across multiple CO problems. **The reason for this result is that single-objective evolution has a smaller search space, making it prone to converging on suboptimal solutions.** In multi-objective evolution, the best algorithm for each problem remains in the Pareto Front, and the best algorithms for different problems vary significantly. Their crossover combinations generate more diverse new algorithms, leading to the discovery of more generic solutions. When using the average improvement ratio as a single objective, the diversity of algorithms within the population is insufficient, resulting in a limited exploration of the algorithm space.

### D.5 OTHER DIVING HEURISTICS IN THE PARETO FRONT

By employing multi-objective evolution, we ultimately obtain a Pareto Front consisting of different diving heuristics. In addition to the diving heuristic with cross-benchmark generalization ability mentioned in Table 3, the Pareto Front retains the best-performing algorithms on each benchmark (as they are not dominated by other algorithms). The results in Table 13 show that the algorithms from the multi-objective evolutionary Pareto Front are competitive with those from single-objective evolution on a specific CO problem. This implies that through **one** multi-objective evolutionary process, users can select algorithms from the Pareto Front based on their practical needs—whether they require algorithms with cross-benchmark generalization ability or those that perform exceptionally well on a single CO problem. This significantly enhances the practical value of LLM4Solver.

Table 13: Compare the best-performing algorithms on each benchmark in the multi-objective evolutionary Pareto Front with single objective evolution. The average relative primal gap with standard error.

| Benchmark | SOEA | MOEA |
|---|---|---|
| Setcover | 3.36(0.25) | 3.41(0.28) |
| Cauctions | 1.83(0.16) | 2.31(0.20) |
| Facilities | 0.65(0.03) | 0.75(0.03) |
| Indset | 0.84(0.07) | 1.03(0.09) |

## E PROMPTS

Prompts are key to whether LLMs can generate effective diving heuristics. We divide the prompts into two groups: 1) background prompts, which provide sufficient background knowledge about MILP problems and diving heuristics; 2) task-specific prompts, which offer detailed instructions for the LLM's specific operations (initialization, crossover, and mutation) to generate new algorithms.

As shown in Figure 5, background prompts contain *Introduction of MILP*, *Definition of MILP*, *Primal Heuristics*, *Diving Heuristics*, *Pseudo-code of Generic Diving* and *Background Instruction*. Together they provide enough background knowledge of diving heuristics for the downstream tasks. Also, shown in Figure 6, task-specific prompts contain the *Task Prompt*, *Features Description*, *In-out Format Description* and *Inspiring Instruction*. By combining the background and task-specific prompts, we get the total prompts for each operator. LLMs take these prompts as input and output

one diving score function named "myheurdiving". An example of the code generated by LLMs is shown in Appendix G.

## F  Non-dominated Sorting and Crowding Distance Sorting

In multi-objective evolution, both parent selection and survivor selection require comparisons between two or more individuals. Therefore, an appropriate method for individual comparison is crucial. We employ Non-dominated Sorting and Crowding Distance Sorting to compare individuals. In the Non-dominated Sorting phase, each individual $s$ is assigned a $rank$, where $rank = 1$ indicates that $s$ is not dominated by any other individual in the population, while a $rank = n + 1$ indicates that $s$ is only dominated by individuals with $rank \leq n$. Thus, a lower rank is preferred.

If two individuals have the same $rank$, we then compare their crowding distance, which measures the distance of $s$ from other individuals $s'$ in the population. To encourage diversity within the population, a larger crowding distance is favored.

In binary tournament selection, we randomly select two individuals from the population at a time and choose the better one after comparison. After $l$ selections, we obtain $l$ parent individuals. In elitism survivor selection, we compare all individuals in the population and select the top $N$ individuals to form the next generation.

---

**Algorithm 3** Non-dominated Sorting

**Input:**  A population $P = \{s_1, s_2, ..., s_N\}$

1: Initial $k = 1, Q = \emptyset$
2: **while** $P \neq \emptyset$ **do**
3:   **for** each $s_i \in P$ **do**
4:     **if** $s_i$ is not dominated by any $s_j$ in $P$ **then**
5:       $rank(s_i) = k$
6:       $Q = Q \cup \{s_i\}$
7:     **end if**
8:   **end for**
9:   $P = P \backslash Q$
10:   $k = k + 1$
11: **end while**

---

**Algorithm 4** Crowding Distance Assignment

**Input:**  $Q = \{s_1, s_2, ..., s_{num}\}$

1: for each j, set $Q[j]_{distance} = 0$
2: **for** each objective $f_i$ **do**
3:   $Q = sort(Q, f_i)$
4:   $Q[1]_{distance} = \inf$
5:   $Q[num]_{distance} = \inf$
6:   **for** $j = 2$ to $num - 1$ **do**
7:     $Q[j]_{distance} = Q[j]_{distance} + \frac{f_i(Q[j+1]) - f_i(Q[j-1])}{f_{i,max} - f_{i,min}}$
8:   **end for**
9: **end for**

---

## G  Examples of Disigned Diving Heuristics

We present examples of diving heuristics designed by LLM4Solver in Figures 7 - 11, which can be directly integrated into SCIP to reproduce experimental results. Notably, before generating the code, the LLM produces a description that guides and explains the execution logic of the code, aiding users in understanding it. Furthermore, the resulting diving heuristics can be categorized into two styles: 1) one resembles linear regression, where inputs are linearly combined and the LLM and evolution adjust the weights for each feature (e.g., GPT-3.5-turbo-16k for Setcover); 2) the other employs complex logical controls and computations to enhance performance (e.g., Claude-3.5-Sonnet for MOEA). These complex logical controls, parameter selections, and computational methods can better utilize existing features, offering greater representational power and practical value compared to simple mathematical expressions(Kuang et al., 2024a) and neural network parameters(Paulus & Krause, 2024). Moreover, this suggests to solver designers that they can not only explore new features as inputs but also leverage existing features to adjust computational methods and parameters, leading to better algorithmic performance.

Figure 5: The background prompts include the <Introduction of MILPs>, <Definition of MILP>, <Primal Heuristics>, <Diving Heuristics>, <Pseudo-code of Generic Diving> and <Background Instruction>. Add them together to get the total background prompts.

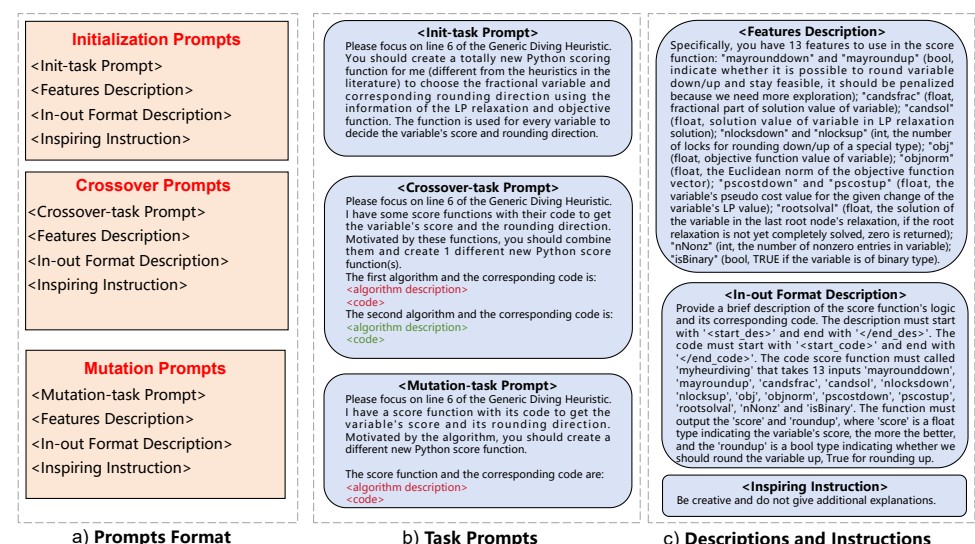

a) **Prompts Format**  b) **Task Prompts**  c) **Descriptions and Instructions**

Figure 6: Task-specific prompts. It is needed to get the parent algorithms to generate the offspring for crossover and mutation prompts.

**Description:**
The new score function aims to prioritize variables with higher potential for improving the objective function value and those that are closer to an integer value. It penalizes variables with limited rounding options and higher locking constraints. The function assigns a higher score to variables with lower fractional parts and higher pseudo cost values, while also considering the number of non-zero entries and binary nature of the variable.

**Code:**
```python
def myheurdiving(mayrounddown, mayroundup, candsfrac, candsol,
nlocksdown, nlocksup, obj, objnorm, pscostdown, pscostup, rootsolval,
nNonz, isBinary):
    # determine roundup
    if mayroundup and not mayrounddown:
        roundup = True
    elif mayrounddown and not mayroundup:
        roundup = False
    else:
        roundup = True  # default to round up if both options are available

    # calculate score
    score = objnorm * (candsfrac * 2 - abs(1 - candsfrac))  # prioritize
variables closer to an integer
    score += (1 / (1 + nlocksdown + nlocksup))  # penalize variables with
locking constraints
    score += candsol * (1 + pscostup + pscostdown)  # prioritize variables
with higher potential for improving objective
    score += 1 / (1 + rootsolval)  # consider the solution value in the last
root node's relaxation

    if isBinary:
        score *= nNonz  # take into account the number of non-zero entries
in binary variables

    return score, roundup
```

**(a) GPT-3.5-turbo**

**Description:**
The new score function is designed to prioritize variables that have a higher probability of rounding up and contribute more to improving the objective function. The score is determined by a combination of several factors: the fractional part of the solution value, the number of locks for rounding down and rounding up, the normalized objective function value, the pseudo cost for rounding up, the number of nonzero entries in the variable, and whether the variable is binary. The score is penalized if it is not possible to round the variable down or if rounding up is not beneficial. The rounding direction is determined based on whether rounding up is more advantageous or rounding down is not allowed.

**Code:**
```python
def myheurdiving(mayrounddown, mayroundup, candsfrac, candsol,
nlocksdown, nlocksup, obj, objnorm, pscostdown, pscostup, rootsolval,
nNonz, isBinary):
    score = 0
    if mayroundup and not mayrounddown:
        score -= 1

    score += candsfrac * 10

    if obj > 0:
        score += objnorm

    score += nlocksup * 0.5

    if pscostup != 0:
        score += 1 / pscostup

    score += nNonz * 0.1

    if isBinary and obj > 0:
        score += 1

    if score < 0:
        score = 0
    roundup = True if score > 0 else False

    return score, roundup
```

**(b) GPT-3.5-turbo-16k**

**Description:**
The scoring function considers both the objective function impact and the feasibility impact when deciding the variable to round and the rounding direction. If rounding down is feasible, it prioritizes variables with larger objective coefficients, smaller fractional parts, and lower pseudo-costs for rounding down. If rounding up is feasible, it prioritizes variables with smaller objective coefficients, larger fractional parts, and lower pseudo-costs for rounding up. The function also penalizes variables with more locks (nlocksdown or nlocksup) to encourage exploration.

**Code:**
```python
def myheurdiving(mayrounddown, mayroundup, candsfrac, candsol,
nlocksdown, nlocksup, obj, objnorm, pscostdown, pscostup, rootsolval,
nNonz, isBinary):
    if mayrounddown:
        scoredown = obj / objnorm - candsfrac - pscostdown / 1000 -
nlocksdown / 100
    else:
        scoredown = -1e9

    if mayroundup:
        scoreup = -obj / objnorm + candsfrac - pscostup / 1000 - nlocksup /
100
    else:
        scoreup = -1e9

    if scoredown > scoreup:
        score = scoredown
        roundup = False
    else:
        score = scoreup
        roundup = True

    return score, roundup
```

**(c) Claude-3.5-Sonnet**

**Description:**
The new `myheurdiving` function is designed to streamline decision-making in MILP diving heuristics by balancing multiple aspects of variables and their situational contexts. It leverages both the fractional part of the variable's solution and its proximity to the root relaxation, allowing for strategic rounding decisions that account for potential future solutions. The function also integrates constraint-based penalties, factoring in the difficulty of locking constraints to balance feasibility against objective optimization. Special treatment for binary variables expedites rounding, while the overall scoring system synthesizes urgency, constraint influence, and exploration needs, aiming for an efficient and directed solution path.

**Code:**
```python
def myheurdiving(mayrounddown, mayroundup, candsfrac, candsol,
nlocksdown, nlocksup, obj, objnorm, pscostdown, pscostup, rootsolval,
nNonz, isBinary):
    # Determine rounding direction based on various factors
    if isBinary:
        roundup = candsol > 0.5
    else:
        if mayrounddown and mayroundup:
            roundup = pscostup < pscostdown if candsfrac < 0.5 else True
        elif mayrounddown:
            roundup = False
        elif mayroundup:
            roundup = True
        else:
            roundup = pscostup < pscostdown
    # Proximity to integer and penalty for rounding direction
    distance_to_integer = min(abs(candsol - int(candsol)), abs(candsol -
(int(candsol) + 1)))
    feasibility_penalty = (nlocksdown if not roundup else nlocksup) * 0.1
    # Constraint influence scaled by normalized objective
    constraint_influence = nNonz / (1 + objnorm + 0.01 * obj)
    # Final score combines distance, influence, and penalties
    score = (1 - distance_to_integer) * constraint_influence -
(feasibility_penalty + (pscostup if roundup else pscostdown))
    if isBinary:
        score *= 1.25  # Prioritize binary rounding
    return score, roundup
```

**(d) GPT-4**

Figure 7: The description and code designed by different LLMs with LLM4Solver trained on Setcover.

**Description:**
This new score function aims to prioritize variables that have a balanced solution value close to 0.5 while considering the potential for exploration by penalizing rounding opportunities. It incorporates the objective function value, pseudo costs, number of locks, and binary variable characteristics to determine the final score and rounding direction.

**Code:**
```python
def  myheurdiving(mayrounddown, mayroundup, candsfrac, candsol,
nlocksdown, nlocksup, obj, objnorm, pscostdown,  pscostup, rootsolval,
nNonz, isBinary):
    score = candsfrac  # Base score from the fractional part

    penalty = 0.4 * (mayrounddown + mayroundup)  # Penalize rounding
potential for exploration
    score -= penalty

    proximity = (0.5 - abs(candsfrac - 0.5)) * 1.8  # Preference for proximity
to 0.5
    score += proximity

    if mayroundup:
        score += (obj * pscostup) / (objnorm + 1e-10)  # Adjust score based
on objective function value and pseudo cost
    if mayrounddown:
        score -= (obj * pscostdown) / (objnorm + 1e-10)

    score += 0.3 * nNonz  # Reward for more nonzero entries

    if isBinary:
        binary_adjustment = 1.8 if candsol > 0.5 else -1.8  # Adjust score for
binary variables
        score += binary_adjustment

    roundup = (score > 0)  # Determine rounding direction based on
accumulated score
    return score, roundup
```

**(a) GPT-3.5-turbo**

**Description:**
The new score function calculates the variable's score based on a combination of 13 features. It penalizes variables that cannot be rounded down or up by deducting a large negative value from the score. It considers the fractional part of the variable's solution value and gives more weight to variables that are closer to an integer value. The score also takes into account the number of locks for rounding down and up, with a higher weight given to variables that have been extensively explored. The objective function value and objective function norm are used to evaluate the importance of the variable in the optimization. Pseudo cost values are considered to determine the direction of rounding. Finally, variables that are binary are treated differently, with the rounding direction based on the number of locks for rounding down or up. The new score function calculates a composite score based on these factors and determines the rounding direction based on the score value.

**Code:**
```python
def  myheurdiving(mayrounddown, mayroundup, candsfrac, candsol,
nlocksdown, nlocksup, obj, objnorm, pscostdown,  pscostup, rootsolval,
nNonz, isBinary):
    score = 0.0
    roundup = False

    if not mayrounddown and not mayroundup:
        score -= 1000000.0

    score += nNonz * (candsfrac + 1) / (nlocksdown + nlocksup + 1)
    score += objnorm * candsol / (max(pscostdown, pscostup) + 0.01)
    score += abs(obj) / (nlocksdown + nlocksup + 1)
    score += rootsolval * (1 + isBinary)

    roundup = True if score > 0 else False

    return score, roundup

    return score, roundup
```

**(b) GPT-3.5-turbo-16k**

**Description:**
The scoring function aims to prioritize variables that have a significant impact on the objective function and can potentially lead to a feasible integral solution. It considers various features to determine the rounding direction and the corresponding score for each variable. The logic is as follows:
1. If the variable is binary, favor the direction that moves the fractional value closer to the corresponding integral value.
2. For non-binary variables, prioritize the direction that has a lower pseudo-cost and fewer locks.
3. Assign a higher score to variables with a larger absolute objective coefficient and a fractional value closer to the midpoint (0.5).
4. Penalize variables that cannot be rounded in either direction by assigning a lower score.

**Code:**
```python
def  myheurdiving(mayrounddown, mayroundup, candsfrac, candsol,
nlocksdown, nlocksup, obj, objnorm, pscostdown,  pscostup, rootsolval,
nNonz, isBinary):
    score = 0.0
    roundup = False

    if isBinary:
        if candsfrac < 0.5:
            roundup = False
        else:
            roundup = True
        score = abs(obj) * (1 - abs(candsfrac - (0 if not roundup else 1)))
    else:
        if pscostdown < pscostup and nlocksdown <= nlocksup:
            roundup = False
        else:
            roundup = True
        score = abs(obj) * (1 - abs(candsfrac - 0.5))

    if not mayrounddown and not mayroundup:
        score *= 0.1

    return score, roundup
```

**(c) Claude-3.5-Sonnet**

**Description:**
The new score function penalizes variables that are easily roundable, have high objective values, or binary variables that are not at extreme values. It prioritizes variables with high fractional parts and low pseudo costs, considering the number of locks, the Euclidean norm of the objective function vector, the solution value in the last root node's relaxation, and the number of nonzero entries in the variable. The function aims to balance exploration and exploitation while taking into account various factors that contribute to the variable's importance in the MILP problem.

**Code:**
```python
def  myheurdiving(mayrounddown, mayroundup, candsfrac, candsol,
nlocksdown, nlocksup, obj, objnorm, pscostdown,  pscostup, rootsolval,
nNonz, isBinary):
    score = 0
    # Penalize easily roundable variables
    if candsfrac < 0.5:
        score -= candsfrac
    if candsfrac == 1:
        score -= 0.2
    if candsfrac == 0:
        score += 0.2
    # Penalize high objective values
    if obj > 0.5:
        score -= 0.5 * obj
    # Prioritize variables with high fractional parts
    score += candsfrac
    # Consider the number of locks
    score -= nlocksdown * 0.05
    score -= nlocksup * 0.05
    # Consider the solution value in the last root node's relaxation
    if rootsolval != 0:
        score += rootsolval
    # Consider the number of non-zero entries in the variable
    if nNonz > 0:
        score += nNonz * 0.1
    roundup = True if score > 0 else False  # Rounding direction based on
the calculated score
    return score, roundup
```

**(d) GPT-4**

Figure 8: The description and code designed by different LLMs with LLM4Solver trained on Cauctions.

**Description:**
The new score function considers the given features to calculate the variable's score for informed rounding decisions. It penalizes limited rounding options, prioritizes variables with high fractional values and lower pseudo costs. It balances exploration and exploitation by factoring in the objective function value, its norm, solution values, the number of locks, nonzero entries, and binary type.
**Code:**
```python
def myheurdiving(mayrounddown, mayroundup, candsfrac, candsol,
nlocksdown, nlocksup, obj, objnorm, pscostdown, pscostup, rootsolval,
nNonz, isBinary):
    penalty = 0
    penalty += 10 if not mayrounddown else 0
    penalty += 10 if not mayroundup else 0

    score = objnorm - penalty + 0.5*candsfrac - (pscostdown + pscostup) -
nlocksdown - nlocksup + obj + abs(rootsolval) - 0.1*nNonz - (0.5*isBinary)

    roundup = score > 0

    return abs(score), roundup
```

**(a) GPT-3.5-turbo**

**Description:**
The new score function's logic is as follows:
1. Start with an initial score of 0.
2. If mayrounddown and mayroundup are both False, penalize the variable by setting the score to a very low value.
3. Calculate the score based on the given features using a combination of arithmetic operations.
4. Adjust the score based on the relative costs of rounding down and rounding up by multiplying it with the ratio of pscostdown to pscostup.
5. Normalize the score based on the magnitude of objnorm by dividing it by the sum of 1 and the square of objnorm.
6. Penalize variables with a larger number of nonzero entries in the variable's solution by subtracting the product of rootsolval and nNonz from the score.
7. Set the rounding direction (roundup) based on whether candsol is greater than or equal to 0.5.
8. Return the final score and rounding direction.
**Code:**
```python
def myheurdiving(mayrounddown, mayroundup, candsfrac, candsol,
nlocksdown, nlocksup, obj, objnorm, pscostdown, pscostup, rootsolval,
nNonz, isBinary):
    score = 0.0
    roundup = False

    if not mayrounddown and not mayroundup:
        score = -1e9
    else:
        score = candsfrac * candsol + (nlocksdown + nlocksup)**2
        score *= pscostdown / pscostup
        score /= (1 + objnorm**2)
        score -= rootsolval * nNonz
        roundup = candsol >= 0.5

    return score, roundup
```

**(b) GPT-3.5-turbo-16k**

**Description:**
The new score function calculates the variable's score based on the given features. It penalizes the variable if it is not possible to round down or up and stay feasible. The rounding direction is determined by comparing the value of 'candsfrac' with 0.5. The score is calculated by combining the features with custom weights. The higher the score, the better. The 'score' and 'roundup' are returned as output.
**Code:**
```python
def myheurdiving(mayrounddown, mayroundup, candsfrac, candsol,
nlocksdown, nlocksup, obj, objnorm, pscostdown, pscostup, rootsolval,
nNonz, isBinary):
    # Define custom weights for each feature
    weights = {
        "mayrounddown": -0.5,
        "mayroundup": -0.5,
        "candsfrac": 1,
        "candsol": 0,
        "nlocksdown": -2,
        "nlocksup": 2,
        "obj": 1,
        "objnorm": 1,
        "pscostdown": -1,
        "pscostup": 1,
        "rootsolval": 0.2,
        "nNonz": 1,
        "isBinary": 0
    }
    score = 0
    # Penalize if not possible to round down/up and stay feasible
    if not mayrounddown or not mayroundup:
        score += weights["mayrounddown"] + weights["mayroundup"]
    # Determine rounding direction
    roundup = True if candsfrac > 0.5 else False
    # Calculate score
    score += weights["candsfrac"] * candsfrac + weights["candsol"] * candsol
+ weights["nlocksdown"] * nlocksdown + weights["nlocksup"] * nlocksup +
weights["obj"] * obj + weights["objnorm"] * objnorm + weights["pscostdown"]
* pscostdown + weights["pscostup"] * pscostup + weights["rootsolval"] *
rootsolval + weights["nNonz"] * nNonz + weights["isBinary"] * (1 if isBinary
else 0)
    return score, roundup
```

**(c) Claude-3.5-Sonnet**

**Description:**
The new score function calculates the variable's score based on the provided features. It penalizes if it's not possible to round the variable down/up and stay feasible. The rounding direction is determined based on the fractional part of the solution value. The calculated score and rounding direction are returned as output.
**Code:**
```python
def myheurdiving(mayrounddown, mayroundup, candsfrac, candsol,
nlocksdown, nlocksup, obj, objnorm, pscostdown, pscostup, rootsolval,
nNonz, isBinary):
    # Penalize if not possible to round down/up and stay feasible
    penalty = 0.1 if mayrounddown or mayroundup else 0

    # Determine rounding direction
    roundup = candsfrac > 0.5

    # Calculate score
    score = (candsfrac + (1 - candsfrac) * candsol + penalty)

    # Extra features
    score += (nlocksdown + nlocksup) * 0.05
    score += (obj / objnorm) * 0.2
    score += (pscostdown + pscostup) * 0.1
    score += (1 - rootsolval) * 0.3
    score += nNonz * 0.05

    if isBinary:
        score += abs(candsfrac) * 1000.0

    return score, roundup
```

**(d) GPT-4**

Figure 9: The description and code designed by different LLMs with LLM4Solver trained on Facilities.

**Description:**
The new score function calculates the variable's score by considering features such as the fractional part of the solution value, LP relaxation solution value, number of locks for rounding, objective function value, Euclidean norm of the objective function vector, pseudo cost values, solution value in the last root node's relaxation, number of non-zero entries, and whether the variable is binary. It penalizes limited rounding possibilities, rewards higher objective values, encourages exploration for feasible solutions, and adjusts the score accordingly based on the specified features.

**Code:**
```python
def myheurdiving(mayrounddown, mayroundup, candsfrac, candsol,
nlocksdown, nlocksup, obj, objnorm, pscostdown, pscostup, rootsolval,
nNonz, isBinary):
    # Determine rounding direction
    roundup = True if candsfrac >= 0.5 else False

    # Calculate score based on specified features
    score = (candsfrac * 0.1) - ((nlocksdown + nlocksup) * 0.2) + (obj * 0.3)
- (objnorm * 0.1) - ((pscostdown + pscostup) * 0.3) + (rootsolval * 0.2) +
(nNonz * 0.1)

    # Penalize limited rounding possibilities
    score -= 0.5 if not mayrounddown else 0
    score -= 0.5 if not mayroundup else 0

    # Reward binary variables
    score += 0.5 if isBinary else 0

    return score, roundup
```

**(a) GPT-3.5-turbo**

**Description:**
The new score function's logic combines the 13 given features to determine the score and rounding direction for each variable. It penalizes variables that cannot be rounded down or up by subtracting 1 from the score. It rewards variables with a higher fractional part of the solution value by adding twice the value to the score. The function also considers the difference between the variable's objective function value and its Euclidean norm, rewarding variables with a higher difference. It penalizes the variable's pseudo cost values for rounding down and up by subtracting 0.1 times each value from the score. It penalizes the number of locks for rounding down and up by subtracting their sum from the score. It rewards binary variables by adding 1 to the score. It penalizes variables with a higher number of nonzero entries by subtracting 0.1 times the number of nonzero entries from the score. Based on the final score, the function determines whether to round the variable up or not by checking if the score is greater than or equal to 0.

**Code:**
```python
def myheurdiving(mayrounddown, mayroundup, candsfrac, candsol,
nlocksdown, nlocksup, obj, objnorm, pscostdown, pscostup, rootsolval,
nNonz, isBinary):
    score = 0.0

    if not mayrounddown:
        score -= 1.0
    if not mayroundup:
        score -= 1.0
    score += candsfrac * 2.0
    score += candsol * 0.5
    score += abs(obj - objnorm) * 0.3
    score -= pscostdown * 0.1
    score -= pscostup * 0.1
    score -= (nlocksdown + nlocksup) * 1.0
    if isBinary:
        score += 1.0
    score -= nNonz * 0.1

    roundup = score >= 0.0

    return score, roundup
```

**(b) GPT-3.5-turbo-16k**

**Description:**
The new score function 'myheurdiving' aims to balance exploration and exploitation by considering objective function values, fractional parts, pseudo-costs, and potential lock-in effects. It prioritizes rounding in the direction that improves the objective function, while promoting exploration by penalizing variables that may lead to infeasibility or lock-in situations. The score is higher for variables with larger objective function coefficients and fractional parts closer to 0.5, promoting better objective improvements and more exploration.

**Code:**
```python
def myheurdiving(mayrounddown, mayroundup, candsfrac, candsol,
nlocksdown, nlocksup, obj, objnorm, pscostdown, pscostup, rootsolval,
nNonz, isBinary):
    # Determine rounding direction
    roundup = abs(candsfrac - 0.5) <= abs(1 - candsfrac - 0.5)

    # Calculate score
    score = abs(obj) / objnorm
    if roundup:
        score *= (1 - candsfrac)
        score -= pscostup * (1 + nlocksup)
        if not mayroundup:
            score *= 0.1  # Penalize potential infeasibility
    else:
        score *= candsfrac
        score -= pscostdown * (1 + nlocksdown)
        if not mayrounddown:
            score *= 0.1  # Penalize potential infeasibility

    # Adjust score for binary variables
    if isBinary:
        score *= 1.5  # Prioritize binary variables

    return score, roundup
```

**(c) Claude-3.5-Sonnet**

**Description:**
The new scoring function, `myheurdiving`, refines and adjusts its methodology to optimize the decision-making process for rounding in MILP scenarios. It evaluates both upward and downward rounding potential by considering the feasibility, direct impact on the objective, and the strategic influence of each rounding decision. The score calculation involves considering penalties for infeasibility and rewards for reducing the objective function while incorporating a complexity factor based on the variable's involvement in the model. Special attention is given to binary variables by assigning them additional score points, enhancing their prioritization due to their decisive nature in binary decisions. This function aims to balance between exploring new feasible regions (penalized if not feasible) and exploiting the current position to move towards optimality by dynamically comparing the potential outcomes of rounding decisions.

**Code:**
```python
def myheurdiving(mayrounddown, mayroundup, candsfrac, candsol,
nlocksdown, nlocksup, obj, objnorm, pscostdown, pscostup, rootsolval,
nNonz, isBinary):
    # Calculate penalties for infeasibility
    penalty_down = 2 if not mayrounddown else 1
    penalty_up = 2 if not mayroundup else 1
    # Base score calculations considering penalties
    score_down = (obj * (1 - candsfrac) - pscostdown * penalty_down +
rootsolval * 3) / (nlocksdown + 1)
    score_up = (obj * candsfrac - pscostup * penalty_up + rootsolval * 3) /
(nlocksup + 1)
    # Boost scores for binary variables
    if isBinary:
        score_down += 50
        score_up += 50
    # Decide the best rounding direction
    roundup = score_up > score_down
    # Normalize the score by considering the objective norm and the
model's complexity
    final_score = (score_up if roundup else score_down) / (objnorm *
(nNonz + 1))

    return final_score, roundup
```

**(d) GPT-4**

Figure 10: The description and code designed by different LLMs with LLM4Solver trained on Indset.

**Description:**
The new score function penalizes limited rounding options to encourage exploration, prioritizes variables with high fractional values and low pseudo costs, considers the impact of the objective function value and Euclidean norm, incorporates historical solution values, adjusts for sparsity, and differentiates based on the binary nature of the variable for effective solution search.

**Code:**
```python
def  myheurdiving(mayrounddown, mayroundup, candsfrac, candsol,
nlocksdown, nlocksup, obj, objnorm, pscostdown,  pscostup, rootsolval,
nNonz, isBinary):
    # Initialization
    score = 0.0
    roundup = False
    # Penalize limited rounding options to encourage exploration
    score -= 0.2 if mayrounddown or mayroundup else 0.0
    # Prioritize variables with high fractional values and low pseudo costs
    score += candsfrac
    score += min(1 / (1 + (pscostdown + pscostup)), 1)
    # Consider the impact of objective function value and Euclidean norm
    score += (obj / max(1, objnorm)) * (1 - candsfrac) / (nNonz + 1)
    # Incorporate historical solution values
    score += rootsolval * 0.3
    # Penalize excessive sparsity
    if nNonz < 5:
        score -= 0.2
    # Adjust based on the number of locks for rounding down/up of a
special type
    score -= min(0.2 * (nlocksdown + nlocksup), 0.4)
    # Differentiate based on the binary nature of the variable
    score *= 1.5 if isBinary else 1.0
    # Determine rounding direction based on the score
    if candsfrac > 0.5:
        roundup = True

    return score, roundup
```

**(a) GPT-3.5-turbo**

**Description:**
The new score function's logic aims to prioritize variables that have a high potential for being rounded up while remaining feasible. It penalizes variables that can be rounded down to encourage more exploration and balanced rounding. The score calculation considers features such as the possibility of rounding down or up, the fractional part of the solution value, the solution value itself in the LP relaxation, the number of locks for rounding down and rounding up, the objective function value and its Euclidean norm, the variable's pseudo cost values, the solution value in the last root node's relaxation, the number of nonzero entries in the variable, and whether the variable is binary or not. The score is a weighted sum of these features, assigning higher weights to factors that have a positive impact on rounding up or the objective function value.
The rounding direction is determined based on the comparison between the fractional part of the solution value and the difference between the solution value and its rounded value. If the fractional part is greater than the absolute difference, it suggests that rounding up is more preferable. Therefore, the 'roundup' variable is set to True if the score indicates that rounding up is better, and False otherwise.

**Code:**
```python
def  myheurdiving(mayrounddown, mayroundup, candsfrac, candsol,
nlocksdown, nlocksup, obj, objnorm, pscostdown,  pscostup, rootsolval,
nNonz, isBinary):
    score = (mayroundup * 0.2) - (mayrounddown * 0.5) + (candsfrac * 100)
+ (candsol * 10) + (nlocksup * 0.5) + (nlocksdown * 0.3) + (obj * 0.05) +
(objnorm * 0.1) + (pscostup * 0.3) + (pscostdown * 0.2) + (rootsolval * 3) +
(nNonz * 0.3) + (isBinary * 2)
    roundup = candsfrac > abs(candsol - round(candsol))
    return score, roundup
```

**(b) GPT-3.5-turbo-16k**

**Description:**
The new score function combines the factors of fractional value proximity, objective coefficient, pseudo costs, number of locks, binary status, and nonzero entries. It prioritizes variables with fractional values closer to integers, larger objective coefficients, smaller pseudo costs, and fewer locks. The rounding direction is determined by the potential impact on the objective function and the fractional value. The score is adjusted based on the number of nonzero entries to encourage sparsity.

**Code:**
```python
def  myheurdiving(mayrounddown, mayroundup, candsfrac, candsol,
nlocksdown, nlocksup, obj, objnorm, pscostdown,  pscostup, rootsolval,
nNonz, isBinary):
    # Determine rounding direction
    roundup = False
    if isBinary:
        roundup = candsol > 0.5
    else:
        roundup = obj * (1 - candsfrac) < obj * candsfrac - pscostup +
pscostdown

    # Calculate score
    if roundup:
        if mayroundup:
            score = (obj * candsfrac) / ((nNonz + nlocksup + 1) * (pscostup +
1))
        else:
            score = -(obj * candsfrac) / ((nNonz + nlocksup + 1) * (pscostup +
1))
    else:
        if mayrounddown:
            score = (obj * (1 - candsfrac)) / ((nNonz + nlocksdown + 1) *
(pscostdown + 1))
        else:
            score = -(obj * (1 - candsfrac)) / ((nNonz + nlocksdown + 1) *
(pscostdown + 1))

    return score, roundup
```

**(c) Claude-3.5-Sonnet**

**Description:**
The `myheurdiving` function for MILP diving heuristic assigns scores to fractional variables for rounding decisions based on several considerations to ensure effective and informed rounding. It emphasizes exploration, solution quality, and the strategic importance of variables, incorporating factors like feasibility of rounding directions, variable proximity to integers, and influence on the overall objective. The function adjusts scores for both pseudo costs and the number of constraints a variable impacts when rounded up or down, with special attention to binary variables due to their pivotal role in solution feasibility. It also adjusts the score based on the deviation from the root solution, promoting variables closer to initial estimations to maintain solution stability.

**Code:**
```python
def  myheurdiving(mayrounddown, mayroundup, candsfrac, candsol,
nlocksdown, nlocksup, obj, objnorm, pscostdown,  pscostup, rootsolval,
nNonz, isBinary):
    score = 0.0
    roundup = candsfrac >= 0.5  # Determines the initial rounding direction
    if mayrounddown and mayroundup:
        score -= 0.3
    # Adding proximity to the nearest integer score
    proximity = 20 * (1 - abs(candsfrac - 0.5))
    score += proximity if roundup else -proximity
    # Influence on the objective function and solution quality
    score += 10 * candsol  # Reward higher solution values
    score += (obj / objnorm) * 3  # Normalize contribution to objective
    # Adjusting for pseudo costs
    score -= 0.7 * (pscostup if roundup else pscostdown)
    # Constraint influence through locks
    score -= 4 * (nlocksup if roundup else nlocksdown)
    # Adding a score for the number of nonzero entries to promote
constraints satisfaction
    score += 3 * nNonz
    if isBinary:
        score *= 1.5  # Increase importance for binary variables
    root_discrepancy = abs(candsol - rootsolval) * 7
    score += root_discrepancy if roundup else -root_discrepancy
    return score, roundup
```

**(d) GPT-4**

Figure 11: The description and code designed by different LLMs with LLM4Solver and MOEA.

