# OpenReview forum: "LLM4Solver: Large Language Model for Efficient Algorithm Design of Combinatorial Optimization Solver"
_ICLR.cc/2025/Conference — ICLR 2025 Conference Withdrawn Submission_

### Official Review · Reviewer_SPw2 · 2024-10-19

**Soundness:** 2
**Presentation:** 2
**Contribution:** 2
**Rating:** 3
**Confidence:** 5

**Summary:**

This manuscript proposes an LLM-based method to generate heuristics for solving Combinatorial Optimization Problems (COPs). The main contribution lies in incorporating the multi-objective fitness function to evaluate the performance of the generated heuristics.

**Strengths:**

S1: This manuscript incorporates the multi-objective fitness function to evaluate the performance of the generated heuristics.

S2: This manuscript demonstrates the performance of the proposed LLM4Solver under different LLMs.

**Weaknesses:**

W1: The novelty of the manuscript is limited. There are several existing LLM-based heuristic generation methods [1-3] that employ Initialization, Crossover, and Mutation promptings to generate heuristics. The only innovation of the proposed method seems to incorporate the multi-objective fitness function.

W2: The description of the method is vague and the use of mathematical symbols is repetitive. For example, on Page 4, $n$ denotes the number of objectives. In Eq. (2), $n = 1, 2, ...,N$. In addition, in Eq. (1), $N_{ins}$ denotes the number of instances used for fitness evaluation. In Algorithm 2, $M$ denotes the number of instances. On Page 16, $r=1$. In Algorithm 2, $k = 1, 2, … ,r$.

W3: In Section 4, my primary concern is the lack of comparison with other SOTA LLM-based heuristic generation methods, such as [1-3]. Given that the limited innovation of LLM4Solver lies in the multi-objective fitness function, I am hesitant to assume that the proposed LLM4Solver would outperform EoH [2] and ReEvo [3] on single-objective problems.

W4: In the ablation study, the authors assess the effectiveness of components such as crossover and mutation. As I mentioned in W1, the prior studies [1-3] have demonstrated the effectiveness of these components. The authors should instead focus on evaluating the effectiveness of their multi-objective fitness function, which is the main contribution of this manuscript.

W5: As far as I know, generating heuristics with an LLM requires a predefined seed function; however, this manuscript, including its appendix, lacks any mention of such a seed function. The authors should clarify how the seed function is predefined.

W6: Several concepts are not adequately defined, and there are numerous grammatical errors. For instance, the values in brackets in Table 1 are not explained. On Page 3, the phrase "After generations of iteration" is unclear. On Page 6, the sentence "Specifically, we employ elitism survivor selection (Zhou et al., 2019), which always selects the N individuals with the best fitness from the total (r + 1)N ones after crossover and mutation" has grammatical errors. Additionally, on Page 17, this manuscript mistakenly refers to "gpt35" instead of "gpt3.5."

[1] Mathematical discoveries from program search with large language models. Nature, 625:468–475, 2024.

[2]  Evolution of heuristics: Towards efficient automatic algorithm design using large language model. In International Conference on Machine Learning, 2024.

[3]  Large language models as hyper-heuristics for combinatorial optimization, arxiv, 2024.

**Questions:**

Please refer to the weakness section.

**Details Of Ethics Concerns:**

Please refer to the weakness section.

---

### Official Review · Reviewer_A6Gg · 2024-10-31

**Soundness:** 2
**Presentation:** 2
**Contribution:** 2
**Rating:** 3
**Confidence:** 4

**Summary:**

This paper uses LLMs, coupled with multi-objective evolution, to learn diving heuristics for solving MILPs. The evolution algorithm is similar to [1] but in the MILP context. The multi-objective part considers the performance on multiple benchmark datasets beyond a single benchmark dataset. The authors show the effectiveness of their algorithm on 6 standard MILP benchmark datasets and 20 instances of MIPLIB.

[1] Liu, Fei, et al. "Evolution of Heuristics: Towards Efficient Automatic Algorithm Design Using Large Language Model." Forty-first International Conference on Machine Learning. 2024.

**Strengths:**

- The performance of the resulting diving rules is competitive on a variety of datasets.
- LLM provides more interpretable diving rules (as code and comments) than previous neural network based methods.
- Detailed ablation studies are performed to show the importance of each component.

**Weaknesses:**

- I find the title too broad and does not reflect the scope of this work. I prefer the author adds phrases along the line of “LLMs for *diving heuristic*” in their title, as this work only considers designing diving heuristics but not other components of MILP solvers.
- I find contribution of the paper a bit weak, as the algorithm pipeline mainly builds on top of [1].
- Limited learning baselines have been compared in the paper. The number of MIPLIB instances evaluated (20) is too small.

[1] Liu, Fei, et al. "Evolution of Heuristics: Towards Efficient Automatic Algorithm Design Using Large Language Model." Forty-first International Conference on Machine Learning. 2024.

**Questions:**

- how is the multi-objective evolution method studied in this work different from papers like [2]? I suggest the authors add a discussion to clarify the novelty of the method.
- Is there anyway to evaluate in general whether the logical descriptions (comments) generated for each diving score function correctly represents the meaning of the score function?
- How different are the generated heuristics on different benchmark datasets? Can you summarize the key difference and provide a potential reasoning why certain heuristics work better on certain datasets?
- How would the learning baseline L2Dive perform if trained under multi-objective (i.e. instead of training only on one dataset, you train the method on multiple datasets)? While I recognize that L2Dive’s source code is not publicly available, I’m concerned that the conclusions of this paper may be a bit weak without principally comparing with any learning methods.

[2] Liu, Fei, et al. "Large language model for multi-objective evolutionary optimization." *arXiv preprint arXiv:2310.12541* (2023).

---

### Official Review · Reviewer_onJF · 2024-11-03

**Soundness:** 2
**Presentation:** 3
**Contribution:** 1
**Rating:** 3
**Confidence:** 4

**Summary:**

The paper proposes a novel framework, LLM4Solver, which uses large language models (LLMs) to automatically design high-quality algorithms for combinatorial optimization (CO) solvers. Specifically, LLM4Solver focuses on improving the diving heuristics of exact CO solvers. By leveraging the prior knowledge embedded in LLMs, the framework generates and optimizes algorithmic components using an evolutionary algorithm. Extensive experiments are conducted to demonstrate the superiority of the proposed method over existing human-designed and learning-based heuristics, particularly in terms of solution quality, solving efficiency, and cross-benchmark generalization.

**Strengths:**

- **Cross-Benchmark Generalization**: The framework shows strong potential for generalizing across different optimization problems, which is a key challenge in CO solver design.
- **Efficiency**: The proposed method demonstrates a high level of efficiency, outperforming existing baselines within a small number of evolutionary iterations.
- **Clear Writing and Detailed Content**: The paper is well-written, with a clear and logical flow that makes it easy to follow the authors' ideas. The methodology is explained in detail, and the step-by-step description of the LLM4Solver framework is thorough, providing a solid understanding of the approach.

**Weaknesses:**

1. **Methodological Innovation**: The core methodological contribution of the paper appears to closely resemble the Evolution of Heuristics (EOH) approach[1], with the main difference being its application to diving heuristics. The paper does not clearly differentiate itself from EOH. A more explicit discussion of the differences, as well as a comparison with EOH in the experiments, would strengthen the contribution.
2. **Lack of Comprehensive Solver Comparison**: The title and abstract emphasize the role of LLMs in enhancing combinatorial optimization solvers, yet the experiments only compare LLM4Solver against SCIP, which is known to be weaker than state-of-the-art solvers like Gurobi and CPLEX. A broader comparison with these solvers is necessary to validate the general claim of improving CO solvers.
3. **Small Problem Instances**: The experiments are conducted on relatively small problem instances, which limits the scope of the evaluation. In some literatures[2][3], approaches have been tested on large-scale mixed-integer linear programming (MILP) problems with millions of variables and constraints. It would be valuable to see how LLM4Solver performs on larger-scale problems to understand its scalability and broader applicability.
4. **Comparison with Learn2Branch**: There are many ways to enhance CO solvers, and Learn2Branch[4][5][6] is a significant direction in this field. The paper uses Learn2Dive, but it does not clearly explain the advantages of Learn2Dive over Learn2Branch, which is a well-established method in CO solver research. A comparison with mainstream Learn2Branch methods would help demonstrate the effectiveness of the proposed approach in enhancing solver performance.
5. **Application to Other CO Solvers**: The paper focuses on improving the SCIP solver's diving heuristic, but it would be interesting to see how the LLM4Solver framework can be applied to other solvers, especially domain-specific ones like Concorde for the traveling salesman problem (TSP). This could provide insights into the general applicability of the proposed method across different types of CO solvers.

[1] Liu F, Xialiang T, Yuan M, et al. Evolution of Heuristics: Towards Efficient Automatic Algorithm Design Using Large Language Model[C]//Forty-first International Conference on Machine Learning. 2024.

[2] Deb K, Pal K. Efficiently solving: A large-scale integer linear program using a customized genetic algorithm[C]//Genetic and Evolutionary Computation–GECCO 2004: Genetic and Evolutionary Computation Conference, Seattle, WA, USA, June 26-30, 2004. Proceedings, Part I. Springer Berlin Heidelberg, 2004: 1054-1065.

[3] Ye H, Wang H, Xu H, et al. Adaptive constraint partition based optimization framework for large-scale integer linear programming (student abstract)[C]//Proceedings of the AAAI Conference on Artificial Intelligence. 2023, 37(13): 16376-16377.

[4] Gasse M, Chételat D, Ferroni N, et al. Exact Combinatorial Optimization with Graph Convolutional Neural Networks Supplementary Materials[J].

[5] Gupta P, Gasse M, Khalil E, et al. Hybrid models for learning to branch[J]. Advances in neural information processing systems, 2020, 33: 18087-18097.

[6] Gupta P, Khalil E B, Chetélat D, et al. Lookback for learning to branch[J]. arXiv preprint arXiv:2206.14987, 2022.

**Questions:**

1. Could the authors clarify the methodological differences between LLM4Solver and EOH, and include experimental comparisons?
2. How does LLM4Solver perform on larger-scale MILP problems, especially those with millions of variables and constraints?
3. Can the authors include a comparison with other combinatorial optimization solvers like Gurobi and CPLEX in the experiments?
4. What are the specific advantages of Learn2Dive over Learn2Branch, and could the authors include a direct comparison with standard Learn2Branch methods?

---

### Official Review · Reviewer_xioJ · 2024-11-04

**Soundness:** 2
**Presentation:** 3
**Contribution:** 2
**Rating:** 5
**Confidence:** 3

**Summary:**

This paper proposes a framework for the automatic generation of high-quality algorithms to solve MILP, leveraging pre-trained Large Language Models (LLMs) as algorithm generators. The framework explores the code search space with a derivative-free evolutionary optimizer named LLM4Solver. Experiments demonstrate that the heuristics generated by LLM4Solver outperform existing state-of-the-art (SOTA) human-designed and learning-based algorithms in terms of solution accuracy, while also showing strong generalization capabilities across different problem settings.

**Strengths:**

1. Combination of evolutionary algorithm and multi-objective fitness evaluation to improve the solution performance and generalization ability.
2. Impressive performance under different scenarios.
3. Improves both solution quality and solving time efficiency.

**Weaknesses:**

1. As this paper is also an evolutionary algorithm-based method, it doesn't have a comparison to previous related papers like EoH by Liu et al. (2024) and ReEvo by Ye et al. (2024).
2. Limited novelty as the main ideas of LLM4Solver has implemented in previous works for solving COP.

**Questions:**

1. As this method is highly related to Evolutionary Algorithm, comparing with other EA based methods, like EoH by Liu et al. (2024) and ReEvo by Ye et al. (2024), does your method outperforms theirs?
2. Could this LLM4Solver method be able to solve larger sized problems? Will its performance decrease when solving large-scale MIP problems?
3. Can LLM4Solver with SOEA, trained on one benchmark be generalized to other problems/benchmarks?

---

### Official Review · Reviewer_FKyv · 2024-11-05

**Soundness:** 2
**Presentation:** 3
**Contribution:** 2
**Rating:** 3
**Confidence:** 4

**Summary:**

The paper proposes a framework that leverages LLMs to assist in algorithm design for combinatorial optimization problems. While traditional CO solvers often demand significant domain expertise and can be challenging to optimize due to the complexity of the search space, LLM4Solver attempts to address these issues by using an LLM to generate algorithmic components in a programming language space, which are then refined through evolutionary algorithms. The framework is specifically applied to Mixed Integer Linear Programs, where it claims to surpass state-of-the-art human-designed and learning-based heuristics in terms of solution quality and efficiency across various benchmarks.

However, despite these claims, the novelty of the approach may be limited, as similar uses of LLMs for algorithm design and optimization have already been explored in the literature. While LLM4Solver achieves certain performance within a few iterations, the experiments are relatively limited in scope, and the paper does not fully investigate critical aspects such as prompt sensitivity and the comparison to existing evolutionary and neural network-based methods. Additionally, while the code generated by the LLM is interpretable, further exploration is needed to assess whether this truly adds value in practice.

**Strengths:**

The paper introduces a novel use of LLMs in combinatorial optimization, integrating them with evolutionary algorithms to generate and refine algorithmic components. This approach shifts from traditional symbolic or neural network-based methods, offering potential benefits in code interpretability and adaptability. The method demonstrates good efficiency, achieving strong performance with only a few iterations, which could be valuable in resource-constrained environments. However, while the results are promising, the novelty of the approach is limited by existing similar works, and the experiments need to be expanded to fully assess its comparative advantages.

**Weaknesses:**

1.	While several LLM-based optimization works have emerged in recent years, I’ve listed a few key papers below [1]-[6]. The current paper primarily compares its approach to the work of Kuang et al. in both the motivation and experimental sections. However, the idea of using LLMs to prompt code for optimization does not seem particularly novel. The authors should expand their comparisons to include these additional works, both theoretically and experimentally.

2.	Additionally, there are various methods that focus on the symbolic discovery of new operators (without using LLMs). It would be beneficial for the authors to compare their approach with these methods to discuss the advantages and disadvantages of leveraging large models in this context.

3.	Regarding the initialization, crossover, and mutation steps, what clear advantages does the LLM offer over traditional random code block operations? Can you provide specific empirical data or examples to support this claim? This would help in better justifying the proposed method.

4.	The sensitivity of the method to the prompt is a key concern, but this aspect is not thoroughly investigated in the paper. I recommend that the authors explore this issue more deeply.

5.	It would also be helpful to provide a clearer description or example of how the code evolves throughout the iterations of the genetic evolution process. A step-by-step demonstration would make it easier for readers to understand the role of the LLM in the evolution of the code.

6.	The experimental comparison appears limited. The seven human-designed heuristics used in the experiments are relatively simple, and the proposed method is only compared against L2DIVE. Given the rapid development of this field, it is difficult to assess whether the proposed method is truly SOTA. Furthermore, I am curious about how the method performs in comparison to evolutionary methods and neural combinatorial methods. While it may not be necessary to outperform all existing methods, a more comprehensive comparison would provide better insight into the performance of the proposed approach.

7.	The overall method design seems relatively straightforward and appears to rely heavily on the capabilities of LLMs. Many of the observed performance enhancements seem to stem from the LLM’s abilities rather than the intrinsic design of the algorithm itself. A clearer discussion of the algorithm's design and its novelty would be beneficial. Additionally, do the strong reasoning capabilities of models like OpenAI's o1-preview and o1-mini provide additional advantages for LLM4Solver?

8.	The section on multi-objective evolution for cross-benchmark generalization needs further clarification. How does multi-objective evolution contribute to cross-benchmark generalization? Additionally, the experiments conducted to verify this claim should be explained in greater detail.

----

[1] Evolution of heuristics: Towards efficient automatic algorithm design using large language model

[2] LLaMoCo: Instruction tuning of large language models for optimization code generation

[3] LLaMEA: A large language model evolutionary algorithm for automatically generating metaheuristics

[4] ReEvo: Large Language Models as Hyper-Heuristics with Reflective Evolution

[5] Autonomous multi-objective optimization using large language model

[6] OptiMUS: Optimization Modeling Using MIP Solvers and large language models

**Questions:**

See the weakness points.

---

### Note · Authors · 2024-11-25

**Comment:**

I have read and agree with the venue's withdrawal policy on behalf of myself and my co-authors.

**Withdrawal Confirmation:**

I have read and agree with the venue's withdrawal policy on behalf of myself and my co-authors.